# Spirituality, Conspiracy Beliefs, and Use of Complementary Medicine in Vaccine Attitudes: A Cross-Sectional Study in Northern Italy

**DOI:** 10.3390/ijerph22030413

**Published:** 2025-03-11

**Authors:** Verena Barbieri, Stefano Lombardo, Arndt Büssing, Timon Gärtner, Giuliano Piccoliori, Adolf Engl, Christian J. Wiedermann

**Affiliations:** 1Institute of General Practice and Public Health, Claudiana—College of Health Professions, 39100 Bolzano, Italy; 2Provincial Institute for Statistics of the Autonomous Province of Bolzano—South Tyrol (ASTAT), 39100 Bolzano, Italy; 3Professorship Quality of Life, Spirituality and Coping, Faculty of Health, Witten/Herdecke University, Gerhard-Kienle-Weg 4, 58313 Herdecke, Germany

**Keywords:** mandatory vaccination, COVID-19 vaccination, spirituality, trust in institutions, complementary and alternative medicine (CAM), conspiracy thinking

## Abstract

Distinct spiritual attitudes and convictions may promote scepticism towards medical interventions, potentially influencing vaccine attitudes. This study investigated the impact of spirituality and CAM (complementary and alternative medicine) use on perceptions of vaccine harmfulness, with a focus on COVID-19 and mandatory childhood vaccinations. Additionally, it examines whether spirituality indirectly influences vaccine hesitancy through CAM use and beliefs in conspiracies. A cross-sectional probability-based survey was conducted with over 1300 participants from South Tyrol, Italy, in 2023, using the GrAw-7 (Gratitude/Awe) scale as a measure of the experiential aspect of non-religious spirituality. Statistical analysis encompassed Spearman’s correlation and linear regression to assess the associations between spirituality and vaccine perceptions. A mediation model was applied to evaluate the role of spirituality in shaping attitudes towards vaccination. Higher experiential spirituality was associated with increased perceived harmfulness of COVID-19 and mandatory childhood vaccinations. Spirituality as well as perceived harmfulness of COVID-19 vaccination and mandatory childhood vaccination were correlated with age, increased CAM utilisation and conspiracy ideation, while institutional distrust was solely associated with vaccine scepticism but not with spirituality; well-being and altruism were only associated with spirituality. Mediation analysis revealed that experiential spirituality accounted for a modest but significant portion of the influence of CAM use and conspiracy thinking on vaccine perceptions. Experiential spirituality may indirectly influence vaccine perception by affecting CAM use and conspiracy thinking. Public health initiatives should incorporate spiritual beliefs and convictions into communication strategies and present vaccinations as a community responsibility. Engaging spiritual and community leaders can enhance the acceptance of vaccines among spiritually inclined groups.

## 1. Introduction

Research has demonstrated a complex relationship between spirituality and well-being, particularly in the context of physical and mental health. For meaningful cross-cultural comparisons, spirituality, which is often used as a broad multidimensional constraint that may include religious beliefs and practices, is often defined in terms of reverence and connectedness [1]. Studies indicate that individuals with moderate religiosity or spirituality generally rate their physical and mental health as average, with spirituality positively influencing the elderly and those with severe illnesses [2,3]. For instance, spirituality has been found to directly enhance mental well-being in older adults [4], while its integration into patient care supports holistic approaches and respects personal values [5].

Under the different definitions of spirituality, we have found the GrAw-7 (Gratitude/Awe) to be an indicator of the experiential aspect of non-religious spirituality. Thus, it is applicable even to non-religious individuals. This aspect of spirituality is moderate to strongly related to various indicators of religious attitudes and practices (such as perception of the sacred) and intensity practices but also to non-religious meditation practices and mindful awareness [6,7,8]. This is an important aspect when investigating Western spiritual beliefs.

In Italy, before the pandemic, spirituality had also been identified as a crucial factor in trauma recovery, as evidenced in studies following natural disasters such as the Italian earthquake, where spiritual beliefs support resilience [9]. Conversely, individuals struggling to cope with trauma may experience a decline in spiritual conviction, raising questions about how global crises, such as the COVID-19 pandemic, influence spiritual practices. Research during the pandemic has explored the connections between spirituality and public health behaviours, revealing a nuanced relationship between religiosity and adherence to government guidelines [10]. Some studies suggest that strict religious beliefs correlate with misconceptions about the virus and resistance to restrictions, contributing to vaccine hesitancy among individuals with conservative spiritual beliefs [11], while others found an association between specific aspects of non-religious spirituality and related ethics on the one hand and refusal of vaccination [12].

Vaccine hesitancy, defined as delaying or refusing vaccines despite available services, is influenced by factors such as mistrust in healthcare systems, safety concerns, and misinformation [13,14]. This complexity is further influenced by diverse national policies, ranging from mandatory to voluntary vaccination approaches [14,15]. Understanding vaccine acceptance is essential for the development of effective communication strategies to increase its uptake. Research specifically examining spirituality’s role in vaccine scepticism (either as an indicator or a direct cause) is limited, although studies in the Netherlands suggest that spirituality is a consistent predictor of science scepticism, vaccine scepticism, and low trust in science [16]. During the COVID-19 pandemic, religious beliefs and spirituality were associated with greater vaccination scepticism, although this link was attributed to low faith in science rather than conspiracy beliefs [17,18]. Specific forms of spirituality have also been associated with vaccine refusal and endorsement of religious conspiracy theories related to COVID-19 [19,20]. However, spirituality is a complex construct with different layers and aspects; therefore, such findings must be interpreted with caution.

This study builds upon these insights into the association between spirituality and vaccine hesitancy. South Tyrol, situated in Northern Italy at the border with Austria and Switzerland, is a region with diverse linguistic and cultural backgrounds and a history of relatively high vaccine hesitancy [21,22]. The effects of spirituality on vaccine hesitancy are investigated in light of the cultural background of this region, focusing even on interactions with conspiracy thinking and CAM (complementary and alternative medicine) use. Former investigations evidenced low vaccine uptake in South Tyrol, emphasising lack of trust as one of the main reasons for vaccine hesitancy and evidencing the associations between vaccine hesitancy and the use of complementary and alternative medicine [23,24]. Our interest was now to understand whether spirituality is linked to vaccine perceptions and may interact with these already-known factors.

Utilising data from the COVID-19 Snapshot Monitoring (COSMO) survey conducted in South Tyrol 2023, this study incorporated spirituality assessments using the GrAw-7 (Gratitude/Awe) questionnaire [6,7]. During the pandemic, GrAw-7 was found to significantly predict well-being [25], with higher scores correlating with positive mental health outcomes [12]. Vaccine hesitancy has also inclined in the years of the pandemic in South Tyrol [24]; we were interested in understanding whether pandemic-related vaccine perceptions and general vaccine perceptions were affected by spirituality and, if so, whether they were affected in the same way.

This study aimed to elucidate the role of a specific aspect of spirituality in vaccine hesitancy within culturally complex regions, thereby providing insights into public health strategies that acknowledge and respect spiritual diversity.

## 2. Methods

### 2.1. Study Design and Data Collection

This study employed a cross-sectional probability-based mode survey. The Statistical Institute of the Autonomous Province of Bolzano-South Tyrol (ASTAT) recruited a random sample of fully aged citizens of South Tyrol, only excluding persons residing in nursing homes, utilising a stratified sampling strategy by municipality, gender, and age group (18–34, 35–49, 50–64, 65+ years) with the program ‘Surveyselect’ in SAS v9.2 (SAS Institute Inc., Cary, NC, USA). Participants were aged ≥18 years. Participants were invited via letter, including the planned participation date, a link to the online questionnaire (with telephone support) covering demographic, clinical, and socio-behavioural aspects, and a personalised password for use as a pseudo-anonymisation code.

The survey was conducted from 1 until 28 February 2023, in the post pandemic period. Most restrictions had been abolished already in May 2022 in Italy, while in October 2022, the pandemic was declared to be over. COVID-19 vaccination was recommended for groups at risk and persons aged 60 and older. The ASTAT data management was conducted in accordance with the EU General Data Protection Regulation.

About 3839 of the 430,000 adult inhabitants of South Tyrol were invited to participate in the quantitative study using a one-stage random sampling design. The samples were collected independently to ensure anonymity. The sample size was determined based on an expected participation rate of 33%, as observed in previous surveys [26]. Since participation was not obligatory, a participation rate ranging between 30% and 40% was found to be realistic. To adjust for non-response bias and ensure that the sample was representative of the target population, post-stratification weights were calculated to replicate the distribution of the population according to age, gender, citizenship, and residence. Participants were invited via formal correspondence, which included the scheduled participation date; a link to the online questionnaire (with telephone assistance available) encompassing demographic, clinical, and socio-behavioural aspects; and a personalised password to be utilised as a pseudo-anonymisation code.

The questionnaire was an extended version of the COSMO Italy and COSMO Germany surveys [27,28].

The questionnaire was available in the German and Italian language. The German and Italian questions, if not available from COSMO Italy and COSMO Germany surveys or as standardised questionnaire versions, were translated from ASTAT and reviewed for language equivalence by a research group at the Institute for General Practice and Public Health.

### 2.2. Demographics

Sociodemographic variables were used to predict vaccine agreement. Age in years, sex (male, female), educational level (middle school or less, vocational school, high school, university degree or more), Italian citizenship as a dichotomous variable, health profession as a dichotomous variable, chronic diseases as a dichotomous variable, and economic situation in the last three months (better, equal, worse, do not know) were assessed. Sociodemographics for the South Tyrol-specific questions were added, including items for the municipality and for the mother tongue (German, Italian, Ladin, and others).

### 2.3. Spirituality

Spirituality was measured with the GrAw-7 scale [6,7], which comprises 7 questions; the intensity of experiences and perceptions is scored on a 4-point Likert scale (1 = “never”, 2 = “seldom”, 3 = “often”, 4 = “regularly/very often”). The resulting score serves as an indicator of experiential spirituality rather than the cognitive construct of spirituality. It focuses primarily on the experiential aspects of being moved and touched by certain moments and places, nature, mindful periods of pausing in “wonder” and thereby interrupting the routine of daily life concerns, and on the subsequent feelings of gratitude. The scale demonstrated good internal consistency (Cronbach’s alpha = 0.82) and is a valid instrument for assessing the perception of the sacred in one’s life. In the general population, the score, when referred to on a 100% scale, reaches a mean value ± SD of 65.3 ± 19.7 [29]. The instrument is available in the German and Italian versions.

### 2.4. Vaccination and Vaccination Perception

Agreement with COVID-19 vaccination, mandatory childhood vaccination and influenza vaccination was assessed through the following questions: “How many times have you been vaccinated against COVID-19?” (with five response options: “1 time”, “2 times”, “3 times”, “4 or more times, not at all”), “Would you vaccinate your child?” (with three response options: “yes”, “no”, “does not concern me”) and “Did you receive influenza vaccination for the winter 2022/23?” (with three response options: “yes”, “no”, “still have to decide”). The question “Has the pandemic altered your attitude towards vaccination?” was posed with three possible responses: “No”, “Yes, I support it more now”, and “Yes, I support it less now”.

To obtain more detailed information about COVID-19 vaccine hesitancy and general vaccine hesitancy, questions regarding the perceived necessity and harmfulness of vaccinations were included. Questions for perceived unnecessity regarding COVID-19 vaccination/mandatory childhood vaccination were as follows: Vaccination is not necessary because … a. it is not effective; b. natural herd immunity and the immune system is quite enough; c. this disease does not/no longer exist; d. the whole thing is only a profit for the pharmaceutical industry. Questions for perceived harmfulness regarding COVID-19 vaccination/mandatory childhood vaccination were as follows: Vaccination is harmful because … a. long-term risks are not known/risks are bigger than benefits; b. new vaccines pose additional risks in the RNA/are not controlled enough; c. there are doctors who advise against it; d. compulsory corona vaccination with prioritisation of certain groups will lead to major socio-political discussions/bad experiences.

Additionally, the general perceptions of COVID-19 vaccination and childhood vaccination were assessed. All responses were measured on a 6-point Likert scale (1 = completely disagree to 6 = completely agree) [30].

### 2.5. Putative Predictors of Spirituality and Perceived Harmfulness of Vaccination (Independent Variables)

The predictors were derived from the literature and the COSMO questionnaire. Trust in information sources and institutions (health authorities and politics) [31,32] was assessed using a 6-point Likert scale ranging from 1 = “no trust” to 6 = “a lot of trust” (including a seventh option “don’t know”).

Furthermore, this study measured conspiracy beliefs (five questions on a 6-point Likert scale from 1 = “strongly disagree” to 6 = “strongly agree”) [33], focusing on perceived misinformation, mistrust in governments and decision-making, secret activities, and secret organisations. Further instruments were altruism (five questions on a 6-point Likert scale from 1 = “don’t agree at all” to 6 = “completely agree”), CAM consultation within the past 12 months [34,35] and well-being within the last 2 weeks (5 items on a 4 point Likert scale from 3 = “always” until 0 = “never”) [36]. Instruments were available in German- and Italian-validated versions.

### 2.6. Statistical Analysis

Sum scores were calculated for spirituality (range: 0–100), conspiracy theories (range: 5–30), altruism (range: 6–30), well-being (range: 6–30), trust in media (range: 5–30), and trust in institutions (range: 8–48). For trust in institutions and trust in media, the response option “I don’t know” was coded as 3.5 (mean of 1 to 6). A higher sum score indicates greater trust. Additionally, items for harmfulness (range 4–24 and range 4–24, respectively) of vaccination and COVID-19 vaccination were aggregated into scores. Cronbach’s alpha was calculated for all the sum scores to assess reliability. According to Arof et al. [37], Cronbach’s alpha of more than 0.9 was regarded as excellent, of 0.8–0.89 as good and of 0.7–0.79 as acceptable.

The sampling methods ensured that the dataset’s demographic profile accurately reflected the population’s age, gender, local authority, and native language distribution.

The sum scores of spirituality, perceived harmfulness of vaccination, and perceived harmfulness of COVID-19 vaccination were regarded as outcomes/dependent variables.

Metric data are presented as mean ± standard deviation (SD). For metric variables, significant differences between the two groups were calculated using the Mann–Whitney U test. For more than two groups (educational level, mother tongue, economic status, opinion change regarding vaccination, agreement with flu vaccination, agreement with childhood vaccination, times of COVID-19 vaccination), Kruskal–Wallis tests were employed, and Dunn Bonferroni Tests were used for post hoc analyses. In this context, effect sizes were calculated using the SPSS methodology, as described [38]. Group differences were visualised using Boxplots. Spearman’s correlation coefficient measured associations between metric variables [39]. Nominal and ordinal data are presented as absolute numbers and percentages. The chi-squared test was used to examine group differences. To account for multiple tests, the Bonferroni correction was applied.

Stepwise linear regression was employed with the spirituality GrAw-7 sum score, the sum score for perceived harmfulness of COVID-19 vaccination, and the sum score for perceived harmfulness of mandatory childhood vaccination as dependent variables.

Associated predictors were incorporated into the model when they exhibited a significant association with the dependent variable at a minimum of *p* = 0.01. For each step, the significance level for inclusion was 0.05 and for exclusion, 0.1. Significant regression coefficients were presented with 95% confidence intervals (CI). The metric predictors were assessed for linearity using a quadratic term for significance. The vaccination-related query “Did you change your opinion regarding mandatory childhood vaccination due to the pandemic?” was included as a categorical predictor, with “No” serving as the baseline response. The economic situation was incorporated as a categorical variable using the “same economic situation” as the reference point. “Other/more than one language” was introduced as a dichotomous factor in the model.

We examined autocorrelation using Durbin–Watson statistics (of the unweighted model), multicollinearity using the variance inflation factor (VIF), and heteroscedasticity using a scatterplot for the predictive value and studentised residuals. The standardised residuals were evaluated for normality. Model diagnostics were conducted using the Differences in Beta (DFBETA) statistics, Cook’s distance, and leverage points.

Finally, a mediation model was implemented to account for the mediating effect of spirituality on the effect of different predictors on the outcome variable “perceived harmfulness of COVID-19 vaccination”. Mediation modelling was performed using the SPSS macro “PROCESS”, employing Model 4 with covariates.

#### Mediation Model Overview

The mediation analysis, conducted using SPSS PROCESS Model 4, evaluated whether spirituality mediates the effects of independent variables on the perceived harmfulness of COVID-19 vaccination (Figure 1).

Independent variables were selected based on significant associations with the independent variable. Spirituality was incorporated as a mediator if it exhibited significant associations with both the independent variable and the outcome.

A mediation model was established based on the following requirements:Predictors must show significant associations with the perceived harmfulness of the COVID-19 vaccination;Predictors must also show significant associations with spirituality;Both predictors and spirituality significantly predicted the dependent variable in a regression model.

Using classical sample size estimation, assuming a type 1 error of 5% and a power of 95% for an R^2^ = 0.1 significantly different from 0 and 15 predictor variables, a minimum sample size of *n* = 264 would be required. Sample size calculation was performed using G*Power version 3.1.9.4. *p*-values < 0.001 are indicated with ***, <0.01 with **, <0.05, *, and *p*-values ≥ 0.05 are considered non-significant (n.s.). All statistical analyses were performed using the SPSS version 27.0.0.0 (IBM, Armonk, NY, USA) [40].

## 3. Results

### 3.1. Response Rate, Psychometric Properties, and Associations with Vaccination Perceptions

The survey yielded a response rate of 36 per cent, corresponding to 1388 participants. Participants’ age had a mean ± standard deviation (SD) of 50.3 ± 17.48, and 51.0% of the participants were female. 18.1% had an educational status of middle school or less, 28.8% in vocational school, 31.4% in high school and 21.8% in university. A total of 40.5% of the participants were urban residents, and 90.8% had Italian nationality. Regarding native language, 63.1% stated to speak German, 27.1% Italian, 3.7% Ladin, and 6.1% other languages. The instrument measuring spirituality demonstrated good internal consistency, with a Cronbach’s alpha of 0.87. The overall mean score for spirituality was 57.1, with a standard deviation of 18.84, consistent with the results during the second year of the pandemic.

The internal consistency of the sum score for COVID-19 vaccination harmfulness was measured using a Cronbach’s alpha of 0.86, whereas that of mandatory vaccination harmfulness yielded a Cronbach’s alpha of 0.88.

Cronbach’s alpha showed reliability coefficients of 0.83 for conspiracy theories, 0.77 for altruism, 0.85 for well-being, 0.85 for trust in media, and 0.93 for trust in institutions.

Times of COVID-19 vaccination, agreement with flu vaccination, agreement with childhood vaccination were not significantly associated to spirituality (Appendix A).

### 3.2. Sample Characteristics, Spirituality, and Perceived Vaccination Harmfulness

Table 1 presents the sample characteristics and their relationships with spirituality, perceived harmfulness of COVID-19 vaccination, and perceived harmfulness of mandatory childhood vaccination. Assessment of spirituality utilising the GrAw-7 scale revealed significant positive correlations with age, conspiracy thinking, altruism, and well-being and negative correlations with trust in media. Female participants exhibited significantly higher spirituality levels (mean = 60.4) than male (53.7) participants and individuals of Italian nationality displayed marginally higher spirituality levels (57.5) than non-Italian nationals (52.9).

A notable difference was observed in responses to the question, “Did the pandemic change your attitudes towards mandatory childhood vaccination?”. The analysis indicated that respondents who answered, “Yes, I support it more now”, exhibited significantly higher spirituality than those who answered “No” (*p* = 0.003; effect size = 0.09). A significant association between language and spirituality was also found. Post hoc analyses indicated that native Ladin speakers were significantly more spiritual than those with multiple or other mother tongues (*p* = 0.007; effect size = 0.26). German and Italian speakers did not exhibit significant differences in spirituality compared with other language groups.

Participants who consulted CAM providers exhibited significantly higher levels of experiential spirituality (61.86% vs. 55.80%, *p* < 0.001) than those who did not.

Sum scores of perceived harmfulness of COVID-19 vaccination and mandatory childhood vaccination were significantly negatively associated with age and trust in institutions and media and positively associated with well-being and conspiracy thinking. Rural residents, individuals who had consulted a CAM provider within the last 12 months, those who had not consulted a GP within the last 12 months, and individuals who did not trust vaccination personnel demonstrated higher scores of perceived harmfulness for both vaccinations. Furthermore, for both vaccinations, significant differences were observed in the sum of scores across different educational attainment levels and economic circumstances.

Post hoc analyses based on economic circumstances regarding the perceived harmfulness of the COVID-19 vaccination showed significant differences. Individuals reporting a deteriorated economic situation scored significantly higher than those reporting an unchanged economic situation (*p* < 0.001, effect size = 0.16). Moreover, participants who were uncertain about their economic situation scored even higher than those who reported no change (*p* < 0.042, effect size = 0.08). Concerning the overall harmfulness score for mandatory childhood vaccination, all pairwise comparisons yielded statistically significant results. The lowest scores were observed among those reporting improved economic circumstances, with scores progressively increasing as the economic situation worsened. The highest scores were recorded for individuals who were uncertain about their economic situations.

Post hoc analysis of subgroups based on educational background revealed significant disparities in the perceived harmfulness of the COVID-19 vaccination. Individuals with vocational schooling scored significantly higher than those with middle school (*p* = 0.005, effect size = 0.12) and university degrees (*p* < 0.001, effect size = 0.17). Additionally, high school graduates scored higher than university degree holders (*p* = 0.005, effect size = 0.11). Regarding the overall perceived harmfulness of mandatory childhood vaccination, all educational groups, that is middle school (*p* = 0.006, effect size = 0.13), vocational school (*p* < 0.001, effect size = 0.16), and high school (*p* = 0.031, effect size = 0.10), scored significantly higher than university graduates. Age was significantly negatively correlated to a higher educational level (rho = −0.380 ***).

The question, “Has the pandemic changed your view on childhood vaccination?” revealed significant differences in the perceived harmfulness scores for both COVID-19 and mandatory childhood vaccinations. Participants who reported decreased support for vaccination due to the pandemic had significantly higher perceived harmfulness scores for COVID-19 vaccination than participants answering “No” (*p* < 0.001, effect size = 0.34) and participants reporting increased support (*p* < 0.001, effect size = 0.65). Additionally, respondents who answered “No” to a change in their view scored significantly higher than those who indicated increased support (*p* = 0.002, effect size = 0.09). Regarding mandatory childhood vaccination, participants indicating reduced support due to the pandemic had significantly higher harmfulness scores compared with participants answering “No” (*p* < 0.001, effect size = 0.40) and participants indicating more support (*p* < 0.001, effect size = 0.68).

Concerning the sum score of perceived harmfulness of mandatory childhood vaccination, Italian nationals had a marginally yet significantly lower sum score than others. Moreover, a significant difference was observed between the mother tongues. Post-hoc tests for mother tongues showed a higher score for others than for German speakers (*p* = 0.047, effect size = 0.08), as well as for Italian speakers (*p* = 0.001, effect size = 0.16).

### 3.3. Spirituality and Attitudes Toward COVID-19 and Mandatory Childhood Vaccinations

Table 2 presents the perspectives on COVID-19 vaccination and general mandatory childhood immunisation alongside the corresponding spirituality averages. Approximately 60% of respondents indicated agreement or strong agreement with the authorities’ decisions regarding both COVID-19 and mandatory vaccination. These individuals demonstrated marginally, yet statistically significant, higher levels of spirituality than those who did not agree.

A higher proportion of individuals perceived the COVID-19 vaccine as unnecessary and harmful than mandatory childhood vaccinations. Those who deemed the COVID-19 vaccine unnecessary exhibited elevated levels of spirituality compared with others. Participants who viewed either COVID-19 or mandatory childhood vaccinations as harmful displayed significantly higher levels of spirituality than their counterparts.

Regarding perceptions of childhood vaccination in the context of COVID-19, participants who agreed (41%) with the statement “I’m worried about the decline of obligatory vaccination due to the pandemic” demonstrated higher spirituality than those who disagreed.

### 3.4. Predictors of Spirituality and Perceived Harmfulness of COVID-19 and Mandatory Childhood Vaccinations

Spearman’s correlation coefficient analysis revealed a marginally positive association between spirituality and the sum scores for both COVID-19 vaccination harmfulness (0.135; *p* < 0.001), while the correlation with mandatory vaccination harmfulness was less relevant (0.076; *p* = 0.002). Table 3, Table 4 and Table 5 show the results of the regression models explaining Spirituality, perceived harmfulness of COVID-19 vaccination and perceived harmfulness of mandatory childhood vaccination.

The independent factors exhibited weak correlations (r < 0.25).

For spirituality (Table 3), the quadratic terms were not statistically significant, except for well-being, which substituted the linear term. This factor did not alter the model quality and parameters; therefore, we retained the untransformed linear well-being regressor in the model. The model was calculated using *n* = 1369 cases and yielded an overall corrected R^2^ of 0.164. Model diagnostics indicated that the variance inflation factor (VIF) was less than 1.2, suggesting the absence of multicollinearity, and the Durbin-Watson statistic was approximately 2. As several (*n* = 12) outliers with standardised residuals < −3 were identified and residuals were not normally distributed, the model was recalculated by excluding these outliers. Subsequently, the standardised residuals were normally distributed, with a mean of 0. The significant variables remained consistent, and the coefficients changed only marginally. The model without outliers showed a slightly improved fit (R^2^ = 0.182). Model diagnostics using DFBETA statistics, Cook’s distance, and leverage points revealed some outliers; however, detailed analyses indicated that the models were stable and did not change after excluding individual cases. In summary, experiential spirituality was predicted by age, gender, CAM consultation, pandemic-related support for mandatory childhood vaccination, well-being, conspiracy thinking, and altruism.

For the harmfulness score of COVID-19 vaccination (Table 4), quadratic terms were not statistically significant, apart from age, which substituted the linear term. This factor did not alter the model quality and parameters; therefore, we retained the untransformed linear age regressor in the model. The model was calculated using 1369 cases and demonstrated an overall corrected R^2^ of 0.423. Model diagnostics indicated that the variance inflation factor (VIF) was less than 1.2; thus, no multicollinearity was detected, and the Durbin-Watson statistic was approximately 2. Standardised residuals were normally distributed, with a mean of 0. There were *n* = 5 standardised residuals > 3, and *n* = 2 residuals < −3 detected. Upon exclusion of these outliers, the significant variables remained consistent, and the coefficients changed only marginally. The model without outliers exhibited a slightly improved fit (R^2^ = 0.437). Model diagnostics using DFBETA statistics, Cook’s distance, and leverage points revealed some outliers; however, detailed analyses indicated that the models were stable and did not change after excluding individual cases. In summary, the perceived harmfulness of COVID-19 vaccination was predicted by age, CAM consultation, pandemic-related lower support for mandatory childhood vaccination, conspiracy thinking, diminished trust in institutions, and spirituality. Results and the corresponding Table of the regression model to explain the perceived harmfulness of mandatory childhood vaccination have been moved to Appendix A.

### 3.5. Mediation Analysis of Spirituality in Perceived Harmfulness of COVID-19 Vaccination

This section examines the potential mediating role of spirituality in the relationship between specific predictors and the perceived harmfulness of COVID-19 vaccination as a dependent variable.

The final mediation model incorporated age, conspiracy thinking, and CAM use as predictors of spirituality as mediators, as all these variables met the inclusion criteria (Details in (Table 3 and Table 4). Other predictors—trust in institutions, reduced support for childhood vaccination due to the pandemic, and economic uncertainty—were significantly associated with the dependent variable (Table 4) but not associated with spirituality (Table 3) and, therefore, were included as independent predictors not mediated by spirituality.

#### Mediation Results

Table 5 and Figure 2 present the results of the mediation model, including the direct, indirect, and total effects of each predictor.

Experiential spirituality mediates approximately 24% of the positive association between CAM consultation and the perceived harmfulness of COVID-19 vaccination. This indicates that increased spirituality partially accounts for the link between CAM use and perceived vaccination risk. Spirituality mediates about 4.7% of the positive association between conspiracy thinking and the perceived harmfulness of COVID-19 vaccination, showing a minor but significant effect. Spirituality mediates approximately −2.8% of the negative association between age and perceived harmfulness of COVID-19 vaccination, indicating a slight dampening effect.

Predictors such as “lower trust in institutions” and “reduced support for vaccination due to the pandemic” showed direct effects on perceived harmfulness but were not mediated by spirituality. Additionally, economic uncertainty had no significant impact on the outcome variables.

## 4. Discussion

This study found that experiential non-religious spirituality was positively associated with beliefs about the harmfulness of both COVID-19 and mandatory childhood vaccinations, although no significant differences in spirituality levels were observed between vaccinated and unvaccinated individuals across COVID-19, childhood, and flu vaccinations. Thus, higher spiritual awareness was associated with greater perceived harmfulness of COVID-19 vaccination, aligning with heightened scepticism toward vaccine safety and effectiveness, as well as increased use of CAM and a tendency towards conspiracy beliefs. Notably, spirituality did not correlate with trust in healthcare institutions or confidence in vaccination professionals, indicating that this indicator of non-religious spirituality may influence vaccine perceptions through personal health beliefs rather than trust in established medical systems. These findings indicate that while spiritual awareness in terms of wondering awe may not directly deter vaccine uptake, it shapes beliefs about vaccine risks, particularly among those engaged in CAM practices or holding conspiracy-related views. This trend underscores the potential role of spiritual perceptions in vaccine reluctance, shedding light on how individual spiritual perceptions and convictions may influence health-related attitudes, especially in culturally diverse areas such as South Tyrol.

### 4.1. Comparison with Existing Literature

Research shows that many people prefer CAM to conventional treatments due to anticipated benefits, dissatisfaction with standard care, and perceived safety [41,42,43,44,45]. CAM practitioners typically use personalised approaches to vaccination, emphasising patient autonomy and resulting in diverse vaccination practices [43,46]. Furthermore, both anti-vaccination and pro-CAM attitudes correlated with magical health beliefs, indicating a shared cognitive framework [42]. Comparative analyses highlight that CAM-focused pharmacists prioritise individualised patient concerns, whereas biomedically trained pharmacists follow standardised vaccination protocols [41].

Integrating spiritual concerns into health assessments and care is crucial, as evidenced by research [47,48,49]. A reliable and concise survey combining health, fitness, and spirituality has been developed for epidemiological studies. Patients’ desire for spiritual discussions often remains unrecognised by healthcare providers, underlining the need for improved communication in palliative care. Moreover, specialised training is necessary to effectively research the intersection of religion, spirituality, and health, focusing on developing competencies to improve study quality and impact.

The findings presented here are consistent with those of previous studies indicating that spirituality and alternative health practices are frequently associated with vaccine hesitancy, particularly in response to widespread public health crises. Studies from the Netherlands and globally have suggested that individuals with higher spirituality may exhibit greater scepticism toward conventional medical practices and vaccine mandates, partially because of personal beliefs that emphasise natural health [16,17,18,19,20]. For instance, a global study [17] found that higher levels of spirituality (the extent to which participants) considered themselves a spiritual person, the extent to which others considered them a spiritual person and the percentage of participants reporting feeling “spiritual peace and well-being” at least once a week) were correlated with lower COVID-19 vaccination rates, while research from the Czech Republic highlighted associations between spirituality, religious fundamentalism, and conspiracy thinking, contributing to heightened vaccine hesitancy during the pandemic [19]. However, as observed in this study, spirituality does not universally deter vaccination; instead, it often influences perceptions of vaccine safety and necessity rather than outright refusal.

Our findings further corroborate the evidence from prior research demonstrating a strong correlation between CAM use and reduced vaccine confidence. Specifically, higher spirituality was significantly associated with CAM consultations, reinforcing previous findings that individuals who prioritise CAM are more likely to question conventional medical practices, including vaccination [20,24]. Studies have also emphasised that CAM users frequently seek alternative explanations for health and wellness, which may engender scepticism towards mainstream vaccines and heighten concerns about their safety [44,45,50,51]. This aligns with our findings, in which spiritual awareness, in terms of awe perceptions, was associated with a higher perceived harmfulness of COVID-19 vaccines. However, it is important to recognise that spirituality can also provide essential coping mechanisms during pandemics. For example, anthropologists have utilised spiritual practices to effectively manage pandemic-related challenges, highlighting the multifaceted influence of spirituality on health behaviours and attitudes [52].

Research indicates a modest correlation between spirituality and conspiracy thinking, both of which can exacerbate vaccine hesitancy. For instance, a study found that individuals with holistic spiritual beliefs are more susceptible to conspiracy theories, which in turn can lead to scepticism toward vaccinations [53]. Similarly, research has shown that lower health literacy and higher religiosity are associated with increased belief in vaccination-related conspiracies [54]. Furthermore, spirituality demonstrated a modest correlation with conspiracy thinking, a factor recognised to exacerbate vaccine hesitancy. In accordance with previous research, including studies elucidating how conspiracy beliefs and spirituality (conceptualised as belief in spiritual powers that confer protection against diseases or defined differently in systematic reviews) can collectively contribute to diminished trust in scientific recommendations [11,14], this investigation revealed that spiritual awareness or resonance (awe perceptions) mediated certain effects of conspiracy thinking on vaccine perceptions. These findings underscore a broader health perspective among spiritually oriented individuals, where scepticism toward vaccines reflects both reliance on CAM and a tendency to question the motives of public health systems. Addressing these unique concerns in vaccination campaigns may necessitate tailored communication strategies that respect the personal beliefs of spiritually inclined individuals while providing clear, accessible information on vaccine safety and efficacy.

### 4.2. Study Aims and Interpretation of Findings

Awe/gratitude is positively related to well-being, corresponding to samples from Germany, Iran, and Israel prior to and during the pandemic, where awe/gratitude was positively related to well-being (WHO-5), indicating a resource. This investigation sought to ascertain whether this specific aspect of non-religious spirituality could function as a mediating factor in the relationship between various predictors, including CAM utilisation, conspiracy ideation, sociodemographic variables, and vaccine perceptions, with a particular emphasis on COVID-19. Through the identification of spirituality as a significant mediator for both CAM utilisation and beliefs in conspiracies, this research suggests that spirituality may indirectly influence vaccine attitudes by reinforcing personal health beliefs and alternative practices. This indirect influence could potentially explain why individuals with high levels of spirituality are more inclined to perceive vaccines as superfluous or detrimental, particularly during the COVID-19 pandemic [12]. Previous studies have similarly observed the influence of spirituality on health practices, wherein personal beliefs regarding natural health and scepticism towards mainstream medical interventions such as vaccines are prominent [16,17,18,19,20].

Moreover, the finding that spirituality mediates the relationship between conspiracy thinking and perceived vaccine harmfulness underscores the complex mechanisms through which spirituality may influence individual attitudes toward vaccines, particularly in contexts in which scepticism or distrust in medical authorities is prevalent. Research has demonstrated that conspiracy thinking can be intensified in individuals with stronger spiritual beliefs who may question the motives of public health authorities or perceive vaccines as incongruent with their health values [11,14,19]. Notably, while stricter or more conservative spiritual and religious groups, such as segments of the Polish Catholic Church, have shown resistance to vaccination recommendations, many liberal or moderately religious communities have actively encouraged vaccination, promoting it as an act of collective responsibility [55]. Similarly, in Germany, resistance to vaccines was predominantly observed in highly spiritual groups at the extremes, emphasising the need to address vaccine hesitancy by engaging with these outlier populations while distinguishing them from mainstream or liberal spiritual communities [56]. This suggests that initiatives to address vaccine hesitancy in spiritually inclined populations should consider these beliefs and incorporate approaches that respect individual health autonomy while presenting evidence-based information on vaccine safety and efficacy.

Additionally, this study assessed whether sociodemographic factors such as age, gender, and educational level influenced both spirituality and vaccine perceptions. Findings indicated that age and gender were significant predictors of spirituality, with older individuals and females generally reporting higher spirituality scores, which aligns with research showing spirituality’s role in well-being and coping, particularly among older populations [2,3,4]. Interestingly, gender was not associated with vaccine perceptions; thus, addressing spirituality in the vaccination discussion is reasonable for different age groups but not separately for men and women. Finally, there was found a small (2.8%) negative mediating effect of spirituality on the effect of age. Interpretation leads to the conclusion that higher age lowers perceived harmfulness, but with increasing spirituality, this effect gets smaller. However, educational attainment influenced perceptions of vaccine harmfulness without directly affecting spiritual levels. This effect was no longer found in the regression model since age and educational level are negatively associated. These demographic insights underscore the importance of tailoring vaccine communication strategies to address age and educational differences, being aware that older persons generally have a lower educational level and thereby ensuring that public health messaging resonates across diverse backgrounds.

The results highlight the complex influence of spirituality on vaccine attitudes, especially amongst those who utilise CAM and subscribe to conspiracy theories. Although statistical analyses revealed correlations between experiential spirituality and vaccine hesitancy, the practical implications suggest a more intricate interplay of personal health beliefs and non-conventional medical perspectives. The observed mediating effect indicates that spirituality does not operate independently but rather interacts with existing belief systems, intensifying concerns about vaccine safety and effectiveness. Notably, as spirituality was not directly associated with trust in institutions, this implies that spiritual worldviews may influence vaccine attitudes through mechanisms separate from generalised medical scepticism, potentially affecting health behaviours at the level of personal conviction rather than systemic distrust.

These findings underscore the necessity of adapting vaccine communication approaches to effectively reach spiritually oriented communities from a public health standpoint. Whilst spirituality can contribute to resilience and well-being, its connection with alternative health practices may require a more comprehensive strategy for vaccine promotion. Rather than relying solely on biomedical arguments, initiatives that emphasise collective welfare, moral duty, and shared societal principles could improve engagement with these groups. Moreover, collaborating with spiritual and community leaders to convey the safety and significance of vaccination in a manner consistent with spiritual beliefs may help narrow the divide between public health efforts and individuals with strong spiritual inclinations.

### 4.3. Implications for Public Health and Future Research

This research elucidates the potential indirect yet significant influence of a specific form of spiritual awareness resulting in specific attitudes, convictions, and behaviours, particularly among individuals engaged in alternative health practices or those with strong conspiracy beliefs. Public health campaigns aimed at increasing vaccine uptake could potentially benefit from acknowledging spiritual beliefs and adapting their messaging to resonate with individuals who perceive health through a spiritual lens. For instance, presenting vaccination as an act of community care or aligning it with values of personal responsibility and altruism might more effectively engage spiritually inclined populations [57].

Furthermore, this study suggests that integrating spirituality into public health communication could facilitate the reconciliation of traditional medical approaches with alternative health perspectives, recognising holistic health beliefs without dismissing scientific guidance. Engaging spiritual and community leaders may also provide a means to enhance vaccine confidence and acceptance among populations that might otherwise be hesitant owing to spiritual considerations [58].

Additional research is necessary to investigate how specific aspects of spirituality, such as feelings of interconnectedness, respect for natural health, and ethical considerations, relate to vaccine decision-making. Moreover, examining spirituality’s mediating role in vaccine attitudes across various cultural and religious contexts could elucidate the unique pathways through which spirituality influences health behaviours [59]. Building upon this study’s mediation approach, future investigations could further assess how spirituality interacts with other factors, such as trust in the media or socioeconomic position, to identify precise intervention points that can render public health efforts more inclusive and effective.

This research contributes to the global discussion on spirituality and vaccine hesitancy by offering insights into South Tyrol, an area with distinct linguistic, cultural, and healthcare characteristics. Unlike previous studies in Italy, where vaccine reluctance was primarily attributed to mistrust in government and socioeconomic inequalities [60], South Tyrol presents a different scenario due to its widespread use of CAM, strong regional identity, and multilingual population. The results suggest that spirituality, particularly in its connection to CAM usage and conspiracy beliefs, mediates attitudes towards vaccines. This emphasises the need for locally tailored health communication approaches that consider institutional trust, economic factors, and the influence of spiritual worldviews and alternative health beliefs. By examining these aspects in a context where spirituality is not necessarily tied to formal religious affiliation, this study provides a better understanding of vaccine hesitancy beyond prevailing narratives in other European and global settings.

### 4.4. Strengths and Limitations

The utilisation of a mediation model provides a novel analytical approach to elucidate the indirect influence of spirituality on vaccine attitudes, revealing the interactions between CAM use and conspiracy thinking. The GrAw-7 scale facilitated a comprehensive assessment of spiritual awareness in terms of resonating with the sacred in everything around nature experiences, but also related to spiritual/religious practices, attitudes, and behaviours. This specific indicator goes beyond traditional religious contexts, which are particularly relevant in regions characterised by diverse spiritual beliefs. However, other indicators of complex constrictive spirituality were not utilised; whether the findings would have changed remains a matter of speculation.

The focus on South Tyrol offers insights into vaccine attitudes within a unique linguistic and cultural setting, thereby informing public health strategies in complex regions.

Finally, the sampling method did not allow participants to partially answer the questions. Thus, we were able to evaluate only complete questionnaires. Post-stratification weights adjusted for non-response bias. Social desirability bias was accounted for by avoiding sentences having emotionally positive or negative connotations as well as questions that are similar to known slogans or have a desirable or undesirable response. Further, socially accepted, or unaccepted wording was avoided.

Despite these robust findings, this study has several limitations. First, its cross-sectional design precludes longitudinal interpretations, while associations between spirituality and vaccine perceptions were identified but cannot be interpreted in the sense of causal inference; it is not feasible to infer how they change and affect each other over time. Longitudinal research could elucidate the development of these relationships and provide insights into how spirituality and vaccine attitudes evolve over time. Second, the regional focus on South Tyrol may constrain the generalisability of these findings to other populations with different cultural or religious contexts. While the two main languages, German and Italian, did not exhibit any significant associations to spirituality and perceived vaccination harmfulness, the small language group (about 4%) of the Ladin minority exhibited a higher Spirituality and higher perceived harmfulness. Thus, on the one hand, results can be generalised to other Western Countries, but special research on small minorities is needed. The applicability of the results to countries outside of Europe and Northern America is difficult. First, the possibility of consultation of CAM providers may be limited in countries with a lower socioeconomic status, and second, conspiracy thinking and trust in institutions may be different in other cultural and religious realities. Nevertheless, it would be interesting to conduct similar investigations in countries with different socioeconomic conditions and a different cultural background.

## 5. Conclusions

This study underscores the association between experiential spirituality and vaccine perceptions, particularly among individuals with a predisposition towards CAM and those who exhibit conspiracy-related beliefs, highlighting the intricate relationships without implying causation. While spirituality does not appear to directly hinder vaccine uptake, it correlates with heightened perceptions of vaccine risks, particularly within contexts where alternative health practices and distrust in mainstream medical authorities are prevalent. This relationship emphasises that people who score high on spiritual awareness may be indirectly influenced to differ from others with respect to vaccination attitudes by promoting scepticism towards vaccine safety and efficacy. Even if we cannot change the spirituality of people, we know now that we can address information campaigns not only by attempting to build trust but also by addressing information to people preferring CAM use and being spiritual at once. We think that this result is an important insight when focusing vaccine campaigns on vaccine-hesitant persons. These findings emphasise the importance of incorporating spiritual awareness, convictions, and beliefs into public health communication strategies. To address vaccine hesitancy within spiritually inclined populations, public health campaigns could explore framing vaccination in ways that resonate with values such as community care, personal responsibility, or altruism while ensuring that these messages are tailored to the diverse beliefs and perspectives of these groups. Moreover, collaboration with spiritual and community leaders could serve as a strategy to strengthen vaccine acceptance in populations that perceive health through a spiritual perspective. Future research should further explore the interactions between spirituality, CAM use, and beliefs in conspiracies, with an emphasis on understanding how spirituality mediates health behaviours in culturally and religiously diverse contexts. Longitudinal studies and analyses across broader demographic groups are necessary to generalise these findings and refine public health interventions aimed at addressing spirituality-linked vaccine hesitancy.

## Figures and Tables

**Figure 1 ijerph-22-00413-f001:**
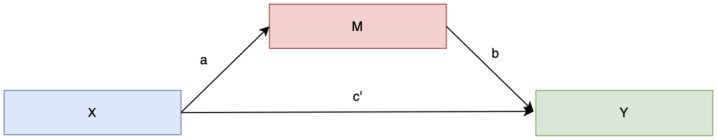
Mediation model 4 (SPSS PROCESS macro). The total effect c of the independent predictor X on the outcome Y is modelled as c = c’+ a × b, with c’ the direct effect of the independent predictor X on Y, the effect of X on the mediator M, and b the effect of the mediator M on outcome Y.

**Figure 2 ijerph-22-00413-f002:**
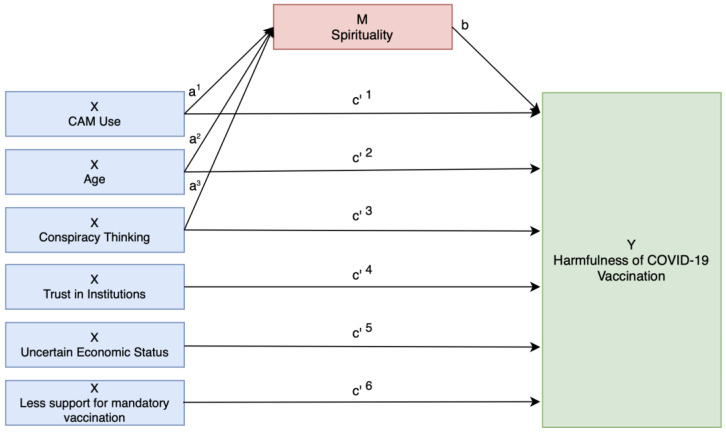
Mediation Model 4 with Covariate Application. Spirituality mediated the effects of CAM use (c^1^ = c’^1^ + a^1^ × b), age (c^2^ = c’^2^ + a^2^ × b), and conspiracy thinking (c^3^ = c’^3^ + a^3^ × b) on the perceived harmfulness of COVID-19 vaccination, while the effects of trust in institutions (c^4^), missing knowledge about the actual economic situation (c^5^), and less support for mandatory childhood vaccination due to the pandemic (c^6^) were not mediated by spirituality. Abbreviations: X, independent predictor; M, mediator; Y, outcome.

**Table 1 ijerph-22-00413-t001:** Characteristics of the sample and associations with spirituality, sum score of perceived harmfulness of COVID-19 vaccination, and sum score of perceived harmfulness of mandatory childhood vaccination.

Characteristics	*n* = 1388	GrAw-7 Spirituality Scale	Sum Score for Harmfulness of COVID-19 Vaccination	Sum Score for Harmfulness of Mandatory Childhood Vaccination
	Mean ± SD	Spearman-rho ^a^	Spearman-rho ^a^	Spearman-rho ^a^
Age	50.3 ± 17.48	0.108 ***	−0.107 ***	−0.058 *
Conspiracy thinking	16.44 ± 6.45	0.122 ***	0.422 ***	0.390 ***
Altruism	20.25 ± 5.24	0.250 ***	0.015 (n.s.)	−0.003 (n.s.)
Well-being	10.91 ± 2.94	0.191 ***	−0.064 *	−0.074 *
Trust in institutions	30.51 ± 9.54	0.035 (n.s.)	−0.516 ***	−0.484 ***
Trust in media	12.29 ± 4.85	−0.063 *	−0.340 ***	−0.247 ***
	%	effect size (*p*-value ^b^) mean	effect size (*p*-value ^b^) mean	effect size (*p*-value ^b^) mean
Gender		0.18 ***	n.s.	n.s.
Female	51.0	60.4	11.6	8.1
Male	49.0	53.7	11.8	8.2
Education		n.s.	0.12 ***	0.11 ***
Middle school or lower	18.1	56.1	11.1	8.2
Vocational school	28.8	55.0	12.0	8.8
High school	31.4	58.2	12.0	8.3
University	21.8	56.6	10.7	7.3
Residence		n.s.	0.075 **	0.054 *
Urban	40.5	57.2	11.1	7.8
Rural	59.5	57.0	12.1	8.4
Citizenship		0.054 *	n.s.	0.055 *
Italian	90.2	57.5	11.7	8.1
Other	9.8	52.9	11.9	8.3
Native Language ^‡^		0.071 *	n.s.	0.094 *
German	63.1	57.4	11.9	8.3
Italian	27.1	57.2	11.2	7.7
Ladin	3.7	61.4	11.5	7.3
Other/more than one	6.1	51.0	11.7	9.1
Working in the health sector		n.s.	n.s.	n.s.
Yes	7.3	59.4	11.8	7.3
No	92.7	56.9	11.7	8.1
Chronic disease(s)		n.s.	n.s.	n.s.
Yes	18.2	58.7	11.5	7.9
No	81.8	56.7	11.7	8.2
Economic situation (last 3 months)		n.s.	0.16 ***	0.17 *
Better	5.0	57.6	11.8	8.2
The same	66.9	57.7	11.1	7.5
Worse	25.2	55.4	13.1	9.6
Don’t know	2.9	48.4	13.6	10.4
CAM consultation		0.12 ***	0.17 ***	0.13 ***
Yes	21.2	61.86	13.6	9.5
No	78.8	55.80	11.2	7.8
GP consultation		n.s.	−0.053 *	0.084 **
Yes	81.7	57.58	11.6	8.0
No	18.3	54.93	12.2	9.1
Trust in vaccination staff		n.s.	0.40 ***	0.383 ***
(Rather) Yes	64.7	57.37	10.1	6.8
(Rather) No	30.8	57.37	14.8	10.7
Don’t know	4.5	51.1	13.5	10.4
Did the pandemic change your opinion about mandatory childhood vaccination?		0.08 **	0.36 ***	0.42 ***
Yes, I support it more now	15.1	60.7	10.0	6.9
Yes, I support it less now	9.6	58.7	18.1	15.3
No	75.3	56.2	11.2	7.5

^a^ *p*-values refer to Spearman’s rank correlation coefficient. ^b^ *p*-values refer to Mann–Whitney U test and Kruskal Wallis test. ^‡^ Mother tongues of South Tyrolean inhabitants. *p*-values: <0.005 = *; <0.001 = **; <0.001 = ***; n.s. = not significant.

**Table 2 ijerph-22-00413-t002:** Attitudes towards COVID-19 and mandatory childhood vaccination.

Category	Question	COVID-19 Vaccination			Mandatory Childhood Vaccination		
		(Rather) Agree %	Spirituality (Mean)	Effect Size (*p*-Value ^a^)	(Rather) Agree %	Spirituality (Mean)	Effect Size (*p*-Value ^a^)
Trust in Authorities	I think that decisions about vaccination made by the public authorities are right	60	56.5	0.05 (**)	60	56.5	0.06 (**)
Perceived Unnecessity	Vaccination is not necessary because …						
	… it is not effective	21	59.2	0.075 (**)	13	57.6	n.s.
	… natural herd immunity/immune system is quite sufficient	21	59.6	0.08 (**)	15	59.13	n.s.
	… this disease does not/no longer exist	5	54.7	n.s.	9	58.01	n.s.
	… the whole thing is only a profit for the pharmaceutical industry	27	59.2	0.08 (**)	18	58.56	n.s.
Perceived Harmfulness	Vaccination is harmful because ...						
	... long-term risks are not known/risk bigger than benefit	46	59.8	0.12 (***)	14	59.3	0.06 (*)
	… new vaccines pose additional risks in the RNA/not controlled enough	24	60.8	0.11 (***)	16	60.0	0.08 (**)
	... there are doctors who advise against it	24	60.0	0.08 (***)	15	60.6	0.07 (**)
	… a compulsory corona vaccination with prioritisation of certain groups will lead to major socio-political discussions/bad experiences	34	61.0	0.15 (***)	15	61.0	0.09 (***)
COVID-19’s Impact on Childhood Vaccination	Considering COVID-19, my views on childhood vaccination are:						
	It is important that my children get the necessary protection	---	---	---	65	57.7	n.s.
	It is important to guarantee heard immunity	---	---	---	64	57.6	n.s.
	I’m worried about the decline of obligatory vaccination due to the pandemic	---	---	---	41	58.6	0.06 (*)
General Childhood Vaccination Attitudes	Despite COVID-19, everybody should be vaccinated according to the national vaccination plan	---	---	---	64	57.6	n.s.
	How serious would the consequences be for the health of your child if you did not do the mandatory childhood vaccination? (*n* = 953)	---	---	---	63	58.1	n.s.
General COVID-19 Vaccination Attitudes	I believe the vaccination can contain the spread of the virus	71	57.34	n.s.	---	---	---
	When all the others are vaccinated against the virus, I don’t need to get vaccinated	10	59.37	n.s.	---	---	---

^a^ *p*-values refer to Spearman’s rank correlation coefficients. *p*-values: <0.005 = *; <0.001 = **; <0.001 = ***; n.s. = not significant.

**Table 3 ijerph-22-00413-t003:** Predictors of spirituality in South Tyrol, Italy, in March 2023 in multivariate linear regression analyses.

Predictors of Spirituality (*n* = 1369)	Correlation R^2^ = 0.164		
	Regression Coefficient b	[95% CI]	*p*-Value
Constant term	14.572	[8.891; 20.254]	<0.001
Age	0.106	[0.053; 0.159]	<0.001
Gender	4.604	[2.722; 6.486]	<0.001
CAM consultation	5.432	[3.166; 7.698]	<0.001
Due to the pandemic, I support mandatory childhood vaccination more now	2.829	[0.240; 5.417]	0.032
Due to the pandemic, I support mandatory childhood vaccination less now			n.s.
Well-being	1.126	[1.440; 0.813]	<0.001
Altruism	0.919	[0.740; 1.098]	<0.001
Conspiracy thinking	0.267	[0.143; 0.410]	<0.001

*p*-values for the significant contribution of independent variables to the model. Abbreviations: CAM, complementary and alternative medicine; CI, confidence interval; n.s. = not significant.

**Table 4 ijerph-22-00413-t004:** Predictors of the harmfulness of COVID-19 vaccination in South Tyrol, Italy, in March 2023 in multivariate linear regression analyses.

Predictors of Harmfulness of COVID-19 Vaccination (*n* = 1369)	Correlation R^2^ = 0.423		
	Regression Coefficient b	[95% CI]	*p*-Value
Constant term	13.100	[11.712; 14.488]	<0.001
Age	−0.025	[−0.038; −0.012]	<0.001
Conspiracy thinking	0.234	[0.196; 0.271]	<0.001
Urban residency			n.s.
Due to the pandemic, I support mandatory childhood vaccination more now			n.s.
Due to the pandemic, I support mandatory childhood vaccination less now	3.832	[3.027; 4.638]	<0.001
CAM consultation	0.707	[0.144; 1.271]	0.014
Better economic situation			n.s.
Worse economic situation			n.s.
Don’t know about the economic situation	1.481	[0.144; 1.271]	0.044
Low educational status			n.s.
Vocational school			n.s.
University degree			n.s.
Trust in institutions	−0.214	[−0.240; −0.188]	<0.001
Trust in vaccination staff			n.s.
Spirituality	0.035	[0.023; 0.047]	<0.001

*p*-values for the significant contribution of independent variables to the model. Abbreviations: CAM, complementary and alternative medicine; CI, confidence interval; n.s. = not significant.

**Table 5 ijerph-22-00413-t005:** Mediation model with mediator (M) spirituality for the independent variable perceived harmfulness of COVID-19 vaccination.

Predictors	C	c’	a × b	%
Age	−0.0282 [−0.0424; −0.0142]	−0.031 ***	0.0029 *	−2.8%
Conspiracy thinking	0.2287 [0.1877; 0.2710]	0.2179 ***	0.0108 *	4.7%
CAM consultation	0.8019 [0.2313; 1.3707]	0.609 *	0.1929 **	24.1%
Support mandatory vaccination less now		3.4919 ***		
Don’t know about my economic situation		n.s.		
Trust in institutions		−0.2207 ***		

*p*-values: <0.005 = *; <0.001 = **; <0.001 = ***; n.s. = not significant.

## Data Availability

The data presented in this study are available upon request from the corresponding author. The data are not publicly available for political reasons because of their conspiracies and ethno-linguistic nature.

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
