# Peer review of "Spirituality, Conspiracy Beliefs, and Use of Complementary Medicine in Vaccine Attitudes: A Cross-Sectional Study in Northern Italy"

_ijerph, 2025, doi:10.3390/ijerph22030413_

Round 1
Reviewer 1 Report
Comments and Suggestions for Authors
This paper studies the impact of spirituality and complementary and alternative medicine on the perceptions of vaccine harmfulness focusing on COVID-19. The paper will be an addition to the large body of studies on vaccine hesitation during COVID-19. However, the paper can provide clarity on certain aspects, particularly on the "sociodemographic factors" that influenced spirituality and vaccine perceptions. How do age and gender, which the research found that they were significant predictors of spirituality, with older individuals and females who were reporting higher spirituality scores, perceive the vaccine? There is no explanation to this question. Similarly, how do educational attainment levels influenced the perceptions of vaccine harmfulness without directly affecting spiritual levels? This is also not explained in details. With this elaboration the paper can be published.
Author Response
Reviewer 1:
Comments 1: This paper studies the impact of spirituality and complementary and alternative medicine on the perceptions of vaccine harmfulness focusing on COVID-19. The paper will be an addition to the large body of studies on vaccine hesitation during COVID-19. However, the paper can provide clarity on certain aspects, particularly on the "sociodemographic factors" that influenced spirituality and vaccine perceptions. How do age and gender, which the research found that they were significant predictors of spirituality, with older individuals and females who were reporting higher spirituality scores, perceive the vaccine? There is no explanation to this question. Similarly, how do educational attainment levels influenced the perceptions of vaccine harmfulness without directly affecting spiritual levels? This is also not explained in details. With this elaboration the paper can be published.
Response 1: Thank you for your comment. We have added in the results at line 319 the sentence “Age was significantly negative correlated to a higher educational level (rho=-0.380***).” In table 1, additionally to the p-values, effects sizes have been added. Thus, associations between demographic variables and spirituality/harmfulness are clearer now. Further, in the discussion, at lines 603-614 we have added additional information about demographic factors and vaccination perceptions
Reviewer 2 Report
Comments and Suggestions for Authors
In the paper by Dr. Barbieri and colleagues, they examined the impact of spirituality and complementary and alternative medicine (CAM) use on perceptions of vaccine harm, particularly regarding COVID-19 and mandatory childhood vaccinations. They reported that institutional distrust is driven by vaccine skepticism. They reported that experiential spirituality may indirectly affect vaccine perception by influencing CAM use and conspiracy thinking. Public health professionals should move forward by incorporating spiritual faith-related organizations into their communication strategies.
This study holds potential interest because the formal analysis appears well-conducted, and the results are noteworthy. However, in its present state, it seems incomplete, with several writing issues, a lack of description of materials and methods, and insufficient discussion.
Introduction
The introduction is thought to be inadequately written and does not sufficiently emphasize the hypothesis of the study.
Lines 13: “the abstract should be a single paragraph and should follow the style of structured abstracts, but without headings” According to the journal rules, it is clearly stated that it should not contain an abstract title.
Lines 15: The sentence that is intended to be expressed should be reviewed again; similar conjunctions have been used too many times in a row.
Line 84: The story pattern in this paragraph is complex and should be simplified. The working hypothesis should be stated more clearly.
Lines 99-109: It would be more appropriate for researchers to specify these questions in the material method section of the manuscript.
Materials and Methods
In the result, there should be a greater focus on practical significance and effect sizes, not just statistical significance. Also, researchers should emphasize more context and interpretation of the findings in the discussion section.
Improve the clarity of some sentences and present the results in a clear and easy-to-understand manner.
Line 117: Data Collection Timeline: While it is stated that data collection occurred in February 2023, more context is needed. What was the COVID-19 status at the time? What were the current vaccination recommendations? This context is critical to interpreting the findings. After all, it is not clear what approach researchers have taken to eliminate the effects of the pandemic at a time when the possible epidemic threats have disappeared, and disinformation about vaccines is circulating due to the influence of other speculators (media).
Line 120: The number of people invited to the study should be stated clearly. It is important how many out of 430 thousand people were reached. The expression more than 4000 is not a suitable expression for this study.
Line 221: Although another study found that over 33% would be considered successful and the study response rate was determined as 39%, the rate of those who did not answer the questions is quite high. This situation should be stated better.
Line 497-Line 503: Reference should be add.
Line 548: this study the regional focus of on South Tyrol, Italy, raises concerns about the generalizability of the findings to other populations with different cultural, religious, and socioeconomic backgrounds. The authors should discuss this limitation further and suggest avenues for future research in more diverse settings.
Line 549: Reference should be add.
Author Response
Reviewer 2:
Comments 1:
Lines 13: “the abstract should be a single paragraph and should follow the style of structured abstracts, but without headings” According to the journal rules, it is clearly stated that it should not contain an abstract title.
Response 1: Thank you, you’re right. Abstract is now a single paragraph
Comments 2: Lines 15: The sentence that is intended to be expressed should be reviewed again; similar conjunctions have been used too many times in a row.
Response 2: sentence changed into “This study investigated the impact of spirituality CAM (complementary and alternative medicine ) use on perceptions of vaccine harmfulness, with a focus on COVID-19 and mandatory childhood vaccinations.”
Comments 3: Line 84: The story pattern in this paragraph is complex and should be simplified. The working hypothesis should be stated more clearly.
Response 3: Now we have written: “This study builds upon these insights into the association between spirituality and vaccine hesitancy. South Tyrol, situated in Northern Italy at the boarder to Austria and Switzerland, is a region with diverse linguistic and cultural backgrounds and a history of relatively high vaccine hesitancy (Kreidl & Morosetti, 2003; Steininger, 2009). The effects of spirituality on vaccine hesitancy are investigated in the light of the cultural background of this region focusing even on interactions with conspiracy thinking and CAM (complementary and alternative medicine) use.”
Comments 4: Lines 99-109: It would be more appropriate for researchers to specify these questions in the material method section of the manuscript.
Response 4: We have eliminated these points from the introduction. The modelling approach is described in the methos section
Comments 5: In the result, there should be a greater focus on practical significance and effect sizes, not just statistical significance. Also, researchers should emphasize more context and interpretation of the findings in the discussion section.
Response 5: We have added effect sizes to all p-values for Man Whitney Test and Kruskal Wallis Test, even in Table 1 and Table 2. Following text was added to the Discussion section:
“The results highlight the complex influence of spirituality on vaccine attitudes, especially amongst those who utilise CAM and subscribe to conspiracy theories. Although statistical analyses revealed correlations between experiential spirituality and vaccine hesitancy, the practical implications suggest a more intricate interplay of personal health beliefs and non-conventional medical perspectives. The observed mediating effect indicates that spirituality does not operate independently but rather interacts with existing belief systems, intensifying concerns about vaccine safety and effectiveness. Notably, as spirituality was not directly associated with trust in institutions, this implies that spiritual worldviews may influence vaccine attitudes through mechanisms separate from generalised medical scepticism, potentially affecting health behaviours at the level of personal conviction rather than systemic distrust.
These findings underscore the necessity of adapting vaccine communication approaches to effectively reach spiritually-oriented communities from a public health standpoint. Whilst spirituality can contribute to resilience and well-being, its connection with alternative health practices may require a more comprehensive strategy for vaccine promotion. Rather than relying solely on biomedical arguments, initiatives that emphasise collective welfare, moral duty, and shared societal principles could improve engagement with these groups. Moreover, collaborating with spiritual and community leaders to convey the safety and significance of vaccination in a manner consistent with spiritual beliefs may help narrow the divide between public health efforts and individuals with strong spiritual inclinations.”
Comments 6: Improve the clarity of some sentences and present the results in a clear and easy-to-understand manner.
Response 6: We have eliminated not significant results from the text and moved all statistical explanations to the Methods section
Comments 7: Line 117: Data Collection Timeline: While it is stated that data collection occurred in February 2023, more context is needed. What was the COVID-19 status at the time? What were the current vaccination recommendations? This context is critical to interpreting the findings. After all, it is not clear what approach researchers have taken to eliminate the effects of the pandemic at a time when the possible epidemic threats have disappeared, and disinformation about vaccines is circulating due to the influence of other speculators (media).
Response 7: We have added information about the Covid-19 status and vaccination recommendations. We have added in the introduction information about former studies indicating that mistrust in institutions and media as well as CAM use were leading causes of vaccine hesitancy during the Pandemic.
Comments 8: Line 120: The number of people invited to the study should be stated clearly. It is important how many out of 430 thousand people were reached. The expression more than 4000 is not a suitable expression for this study.
Response 8: We have changed the sentence to “About 3,839 of the 430,000 adult inhabitants”
Comments 9: Line 221: Although another study found that over 33% would be considered successful and the study response rate was determined as 39%, the rate of those who did not answer the questions is quite high. This situation should be stated better.
Response 9: We have added the sentence “Since participation was not obligatory, a participation rate between 30% and 40% was found to be realistic. To adjust for nonresponse bias and ensure that the sample was representative of the target population, post-stratification weights were calculated to replicate the distribution of the population according to age, gender, citizenship, and residence.”
Comments 10: Line 497-Line 503: Reference should be add.
Response 10: Following text and references were added: In addition to Deml et al., Frawley et al. was added as reference.
Comments 11: Line 548: this study the regional focus of on South Tyrol, Italy, raises concerns about the generalizability of the findings to other populations with different cultural, religious, and socioeconomic backgrounds. The authors should discuss this limitation further and suggest avenues for future research in more diverse settings.
Response 11: We have added the text “While the two main languages German and Italian did not exhibit any significant associations to spirituality and perceived vaccination harmfulness, the small language group (about 4%) of the Ladin minority exhibited a higher Spirituality and higher per-ceived harmfulness. Thus, on the one hand, results can be generalized to other Western Countries, but special research on small minorities is needed. The applicability of the results to countries outside of Europe and Northern America is difficult. First, because the possibility of consultation of CAM providers may be limited in countries with a lower socioeconomic status, second, because conspiracy thinking and trust in institutions may be different in other cultural and religious realities. Nevertheless, it would be interesting to do similar investigations in countries having different socioeconomic conditions and a different cultural background.”
Comment 12: Line 549: Reference should be add.
Response 12: Following text and references were added: “Research indicates a modest correlation between spirituality and conspiracy thinking, both of which can exacerbate vaccine hesitancy. For instance, a study found that individuals with holistic spiritual beliefs are more susceptible to conspiracy theories, which in turn can lead to skepticism toward vaccinations (Jedinger & Siegers, 2024). Similarly, research has shown that lower health literacy and higher religiosity are associated with increased belief in vaccination-related conspiracies (Pavić et al., 2023).”
Reviewer 3 Report
Comments and Suggestions for Authors
I really appreciate your work here. It is vital to public health that all communities, spiritual and otherwise, recognize that cognitive/affective/tribal affiliations do not militate against community responsibilities. Indeed, the irony here appears to be that while spirituality tends to promote interests in the transcendence of self and connectedness, the transcendence and connection are ultimately not humanist in nature. In this regard, spiritually inclined populations may be constituted by selfish individuals. Accordingly, you are right to note that communication strategies must be tailored carefully to the targeted audience. Such sympathetic messaging may help. Alas, other strategies could be employed, such as street epistemology [although this is a longterm strategy]. I will admit that I am not prepared to comment too critically on your research methods, but they seem sound and your assessment of the results appear accurate. I will note, however, that the interests in including spiritual discussions in allopathic care may be unwieldy. Doctors are already pressed for time and expertise. On this score, you may want to check out R. P. Sloan's Blind Faith. I may also note that I have published on the topics of infectious disease and religion and have made arguments to the effect that religion may have been the first stab at public health policy. I encourage you to continue this research project.
Author Response
Reviewer 3:
Comments 1: I really appreciate your work here. It is vital to public health that all communities, spiritual and otherwise, recognize that cognitive/affective/tribal affiliations do not militate against community responsibilities. Indeed, the irony here appears to be that while spirituality tends to promote interests in the transcendence of self and connectedness, the transcendence and connection are ultimately not humanist in nature. In this regard, spiritually inclined populations may be constituted by selfish individuals. Accordingly, you are right to note that communication strategies must be tailored carefully to the targeted audience. Such sympathetic messaging may help. Alas, other strategies could be employed, such as street epistemology [although this is a longterm strategy]. I will admit that I am not prepared to comment too critically on your research methods, but they seem sound and your assessment of the results appear accurate. I will note, however, that the interests in including spiritual discussions in allopathic care may be unwieldy. Doctors are already pressed for time and expertise. On this score, you may want to check out R. P. Sloan's Blind Faith. I may also note that I have published on the topics of infectious disease and religion and have made arguments to the effect that religion may have been the first stab at public health policy. I encourage you to continue this research project.
Response 1: Thank you very much for your appreciation! I searched for the book R. P. Sloan's Blind Faith and will have a look on it. I do not feel ready to cite it in an appropriate way in this paper, but if you want to, you could make a suggestion how to cite it.
Reviewer 4 Report
Comments and Suggestions for Authors
This is a manuscript that provides meaningful information on a topic that can contribute clinically. However, it needs improvement in terms of validity as a scientific study and in writing the paper. Here are somethings I would like you to improve it:
1. You don't need the phrase "Background/Objectives:" in your abstract, because that would result in a double colon, like "Abstract: Background/Objectives:".
2. Although Northern Italy was a region that was severely affected by the early spread of the COVID-19 pandemic, I think the biggest issue in this study is whether the results of this study can be generalized to other parts of the world due to cultural characteristics. In particular, spirituality is a factor that is greatly influenced by culture. It is necessary to clarify how this was controlled in this study.
3. Please clarify what this study contributes beyond what has been done globally and what differentiates it from studies conducted in Southern Italy.
4. Please present your research questions in one paragraph. It is not appropriate to list them in a paper without numbering them and using '•'.
5. I'm skeptical about the significance of these descriptive statistics. Could you clarify that?
6. In multiple regression analysis, you need to have a rational that includes the predictors you have chosen.
7. Examining the mediation model in this kind of research seems somewhat odd.
8. It would be good to present the implications of the results of this study in more detail.
Author Response
Reviewer 4:
Comments 1: You don't need the phrase "Background/Objectives:" in your abstract, because that would result in a double colon, like "Abstract: Background/Objectives:".
Response 1: Thank you, done!
Comments 2. Although Northern Italy was a region that was severely affected by the early spread of the COVID-19 pandemic, I think the biggest issue in this study is whether the results of this study can be generalized to other parts of the world due to cultural characteristics. In particular, spirituality is a factor that is greatly influenced by culture. It is necessary to clarify how this was controlled in this study.
Response 2: We have added the following text in the discussion “While the two main languages German and Italian did not exhibit any significant associations to spirituality and perceived vaccination harmfulness, the small language group (about 4%) of the Ladin minority exhibited a higher Spirituality and higher perceived harmfulness. Thus, on the one hand, results can be generalized to other Western Countries, but special research on small minorities is needed. The applicability of the results to countries outside of Europe and Northern America is difficult. First, because the possibility of consultation of CAM providers may be limited in countries with a lower socioeconomic status, second, because conspiracy thinking and trust in institutions may be different in other cultural and religious realities. Nevertheless, it would be interesting to do similar investigations in countries having different socioeconomic conditions and a different cultural background.”
Comment 3: Please clarify what this study contributes beyond what has been done globally and what differentiates it from studies conducted in Southern Italy.
Response 3: Following text was added to the discussions section: “This research contributes to the global discussion on spirituality and vaccine hesitancy by offering insights into South Tyrol, an area with distinct linguistic, cultural, and healthcare characteristics. Unlike previous studies in Italy, where vaccine reluctance was primarily attributed to mistrust in government and socio-economic inequalities (Bertoncello et al., 2020), South Tyrol presents a different scenario due to its widespread use of CAM, strong regional identity, and multilingual population. The results suggest that spirituality, particularly in its connection to CAM usage and conspiracy beliefs, mediates attitudes towards vaccines. This emphasizes the need for locally tailored health communication approaches that consider institutional trust, economic factors, and the influence of spiritual worldviews and alternative health beliefs. By examining these aspects in a context where spirituality is not necessarily tied to formal religious affiliation, this study provides a better understanding of vaccine hesitancy beyond prevailing narratives in other European and global settings.
Comment 4: Please present your research questions in one paragraph. It is not appropriate to list them in a paper without numbering them and using '•'.
Response 4: We have eliminated the list in the introduction and added “The effects of spirituality on vaccine hesitancy are investigated in the light of the cultural background of this region focusing even on interactions with conspiracy thinking and CAM (complementary and alternative medicine) use.”
Comment 5: I'm skeptical about the significance of these descriptive statistics. Could you clarify that?
Response 5: we have added effect sizes to all significant p-values of Man Whitney and Kruskal Wallis Test to evidence more clearly how variables are associated
Comment 6. In multiple regression analysis, you need to have a rational that includes the predictors you have chosen.
Response 6: Thank you, we have clarified this in the Methods section, changing “Stepwise linear regression was employed with the spirituality GrAw-7 sum score, the sum score for perceived harmfulness of COVID-19 vaccination, and the sum score for perceived harmfulness of mandatory childhood vaccination as dependent variables. For each step, the significance level for inclusion was 0.05 and for exclusion 0.1.”
Comment 7. Examining the mediation model in this kind of research seems somewhat odd.
Response 7: I understand your remark. Spirituality is not a variable, that can be changed by GPs or other stakeholders. Our interest was to examine the effect of spirituality on the complex interplay between vaccine scepticism and other known predictors. Modelling interaction terms did not lead to any significant result and indicating correlations alone seemed us too less. Since we have available a large data set, a more complex modelling approach allows to draw conclusions. Even if we cannot change the spirituality of people, we know now that we can address information campaigns not only by the attempt to build trust, but even by addressing information to persons preferring CAM use and being spiritual at once. We think, this result is an important insight when focusing vaccine campaigns on vaccine hesitant persons. We added in the conclusion this last sentence
Comment 8. It would be good to present the implications of the results of this study in more detail.
Comment 8: Additionally to the text of Comment 3 we added in the discussion: While the two main languages German and Italian did not exhibit any significant associations to spirituality and perceived vaccination harmfulness, the small language group (about 4%) of the Ladin minority exhibited a higher Spirituality and higher per-ceived harmfulness. Thus, on the one hand, results can be generalized to other Western Countries, but special research on small minorities is needed. The applicability of the results to countries outside of Europe and Northern America is difficult. First, because the possibility of consultation of CAM providers may be limited in countries with a lower socioeconomic status, second, because conspiracy thinking and trust in institutions may be different in other cultural and religious realities. Nevertheless, it would be interesting to do similar investigations in countries having different socioeconomic conditions and a different cultural background.”
Reviewer 5 Report
Comments and Suggestions for Authors
Spirituality, Conspiracy Beliefs, and CAM Use in Vaccine Attitudes: A Cross-Sectional Study in Northern Italy
Dear Authors,
The research is worthy of interest, especially in religious societies.
The manuscript is clear and well-constructed ,relevant for the field and presented in well-a structured .
-Introduction:
. The introduction was very good, documented, clear in its objectives, and explained the purpose of the research in detail.
- The main aim: It was completely explained
-Materials and Methods
- Instruments:
The article tools are appropriate
the authors did not mention whether the question was translated from the mother language into Italian.
It would be helpful if this point was explained
-Statistical Analyses
The statistical study was broad, appropriate, and specialized, covering all aspects of the research.
-The statistical study is clear and the tables are appropriate and explain the results.
-Results
It was clearly explained
There are many tables and it is better to delete some of them and explain their contents in text.
-Discussion: The discussion is consistent with the results and well documented.
-References and citations are appropriate.
many thanks
Author Response
Reviewer 5:
Comments 1: The introduction was very good, documented, clear in its objectives, and explained the purpose of the research in detail.
Response 1: Thank you
Comments 2: - The main aim: It was completely explained
Response 2: Thank you!
Comments 3: Materials and Methods:- Instruments:The article tools are appropriate . The authors did not mention whether the question was translated from the mother language into Italian. It would be helpful if this point was explained
Response 3: We added “The questionnaire was available in German and Italian language. The German and Italian versions, if not available from COSMO COSMO Italy and COSMO Germany surveys or as standardized questionnaire versions, were translated from ASTAT and reviewed for language equivalence by a research group at the Institute for General Practice and Public Health.” And in the section “ Putative Predictors of Spirituality (Independent Variables)” we added the sentence “Instruments were available in German and Italian validated version.”
Comments 4: Statistical Analyses. The statistical study was broad, appropriate, and specialized, covering all aspects of the research. The statistical study is clear and the tables are appropriate and explain the results.
Response 4: Thank you
Comments 5: Results: It was clearly explained. There are many tables and it is better to delete some of them and explain their contents in text.
Response 5: Moved Table 5 and the corresponding results to the supplementary file.
Comments 6: Discussion: The discussion is consistent with the results and well documented.
Response 6: Thank you
Comments 7: References and citations are appropriate.
Response 7: Thank you
Reviewer 6 Report
Comments and Suggestions for Authors
The manuscript entitled "Spirituality, Conspiracy Beliefs, and CAM Use in Vaccine Attitudes: A Cross-Sectional Study in Northern Italy" examinates several factors implicated in vaccine attitudes. It is interesting and can give a real scientific contribution in this field. However, it need to be revised in order to improve it, listing the comments below:
-
- first of all, check if citation style used in this manuscript is the same of that required by this journal;
- please, remove the acronym CAM in the manuscript title (line 2) because it cannot be understood by the readers. Write the full form, instead;
- please, give the full form of GrAw-7 (line 20) because it is not clear what is it;
reading the results section in the abstract, I cannot understand the meaning because statistically significant results found in this research lack. Please, add the main statistically significant results and write them more clearly (from the predictive factors to the outcomes); - give the definition of spirituality in this study (line 43) first of all, explaining better its role in human attitudes and the different forms of spirituality (to link it with lines 90-95);
- considering line 85, explain better the reasons for which South Tyrol is an Italian region with high vaccine hesitancy and it is important to evaluate the association spirituality and vaccine hesitancy in this geographical area;
considering line 89, give the full form of GrAw-7; - in line 90 there is an unnecessary space "[...] non-religious spiritu-[...]";
considering lines 96-98, I cannot understand the reasons for which mental health was mentioned if it was wanted to evaluate spirituality and vaccine hesitancy. Please, explain better this choice and its implication in spirituality and vaccine hesitancy; - reading line 117, there is only an inclusion criterion, but it seems to be unlikely. Please, define better inclusion and exclusion criteria for the participation of this study;
- please, add the references of EU General Data Protection Regulation (line 118);
- reading lines 129-134, I cannot understand when the questionnaire was administered (the period of data collection by the questionnaire) nor what the administration methods were (by mailing list? by telephone? by short message system?). Please, clarify this aspects;
- being in South Tyrol, what was the language used for the questionnaire?
- reading lines 136-141, the point scales for the question group are confusing. I strongly suggest dividing the questions by point scales. For example, "Age, gender, educational level on a 4-point scale, [...]" is not clear: age can also have 4-point scale, but it must be defined better how the age classes are chosen, whereas gender can have more than 4-point scale. Whatever scale is chosen, it is important to provide evidence of why that modality was chosen rather than another, giving literature references when appropriate;
- reading lines 139-141, explain better how the "Sociodemographic questions were adapted to the specific South Tyrolean contex";
- reading lines 145-146, I cannot understand what number 2 and 3 of 4-point Likert scale corresponds to. The same is for the point Likert scales of the other questions. Please, clarify these aspects;
- please, add the range of scores for each item (see lines 142-182);
- reading line 148, there are unnecessary spaces in "[...] places / nature [...]";
- as regards lines 179-180, questions about perceived necessity and harmfulness of vaccinations are lacking. Please, add the questions;
- in line 195, Spearman's correlation coefficient is mentioned, but there is a reference about the interpretation. This reference Akoglu H. User's guide to correlation coefficients. Turk J Emerg Med. 2018 Aug 7;18(3):91-93. doi: 10.1016/j.tjem.2018.08.001. PMID: 30191186; PMCID: PMC6107969. can be helpful and added at the end of the sentence;
- please, use the acronym VIF after the full form of variance inflation factor (the acronym must be into brackets for the first time - line 205);
- please, add the full form of the acronym DFBETA;
- considering line 218, add the reference of the statistical software as it is reported in this website: https://www.ibm.com/support/pages/how-cite-ibm-spss-statistics-or-earlier-versions-spss;
- after reading materials and methods section, I cannot understand how the variables were processed, what the independent variables are and the dependend variables. Please, clarify better;
- in line 224, there is a 22 into square brackets. Please, revise it;
- reading lines 238-240, doses of COVID-19 vaccination are mentioned, but they are not included in the materials and methods section. The questions described in the materials and methods section has to be the same of those in the results section, otherwise the readers get confused. Please, revise this aspect paying more attention;
- reading Supplementary Figure S1 (mentioned in line 245), I noted that there is not a relationship between variables, but a descriptive analysis (in detail, median, interquartile range, and outliers) depicted in the box plots. To have a relatioship between variables, a correlation test or regression should be performed, as it is reported in this article: Johannes Ledolter, Oliver W. Gramlich, Randy H. Kardon; Display of Data. Invest. Ophthalmol. Vis. Sci. 2020;61(6):25. https://doi.org/10.1167/iovs.61.6.25. For these reason, I strongly suggest revising the first part of the description of the Figure "Relationship [...] influenza vaccinations.". Moreover, this measure of central tendency is not mentioned in the statistical analysis section and in the last plot there is an incomplete word "Don't kn". Please, revise it;
- reading reading very carefully the results section (more than one), I noted that it is too hard to understand what results are significant for this reasearch. There are descriptions on how the variables were processed (which must be done in an appropriate section of materials and methods) and statistical analyses which are not reported in the relative section. To help in the revision of this section, I try to list how the results could be improved (I hope these comments are useful): - first of all, describe the participants' characteristics (total number of participant, age, gender, and items), paying attention with the acronym (for instance, standard deviation is SD); - lines 227-230 should be inserted in other section which mentions the relationship between variables; - lines 231-232 should be put in the statistical analysis section; - lines 235-236 should be put in the appropriate section of materials and methods on sample methods; - lines 237-247 represent non-statistically significant results of Supplementary Figure S1, so they can be removed from the main manuscript and inserted in such Supplementary Material; - lines 253-255 should include statistically significant results for gender and nationality; - lines 259-261 can be removed because the results are not statistically results; - line 262 mentions additional analysis, which are not reported in the statistical analysis section. Please, revise this aspect; - lines 271-272 must report the statistically significant results (coefficents and other measures); lines 273-276 should be removed because they do not report statistically significant results; - lines 277-285 should include only statistically significant results, removing those non-significant; - lines 286-296 report subgroups analysis considering economic status, which was described in the statistical analysis section. Please, before reporting the results, describe the analysis in the statistical analysis section. Moreover, report only the statistically significant results. Do the same for the lines 297-306; - lines 324-325 should be removed; - lines 327-344 should include only the statistically significant results; - lines 349-353 should be included in the statistical analysis section; lines 353-358 should be included in the appropriate subsection of the materials and methods; - lines 359-360 should be included in the statistical analysis section; - lines 374-376 and 393-396 should be included in the statistical analysis section; the subsection "3.5.1. Mediation Model Overview" should be included in the statistical analysis section;
- line 514 reports an unnecessary bracket;
- lines 549-551 seem to be without references. Please, revise them;
- considering the paragraph "4.4. Strengths and Limitations", common biases for the cross-sectional study must be reported;
- lines 669-672 do not report an approval by a local ethic committee in accordance with the Legislative Decree no. 121 of May 5, 2001. Searching it in the Internet, this source is not available and I cannot check the corrispondence between the Legislative Decree and the Authors' declaration. Although I trust them, I would like to check it. For this reason, I would like to ask the link of the source.
- I hope my comments are intelligible and helpful in improving the quality of the manuscript.
Author Response
Reviewer 6:
Comments 1: first of all, check if citation style used in this manuscript is the same of that required by this journal;
Response 1: Done
Comments 2: please, remove the acronym CAM in the manuscript title (line 2) because it cannot be understood by the readers. Write the full form, instead;
Response 2: Done
Comments 3: please, give the full form of GrAw-7 (line 20) because it is not clear what is it;
Response 3: Done
Comments 4: reading the results section in the abstract, I cannot understand the meaning because statistically significant results found in this research lack. Please, add the main statistically significant results and write them more clearly (from the predictive factors to the outcomes);
Response 4: Done
Comments 5: give the definition of spirituality in this study (line 43) first of all, explaining better its role in human attitudes and the different forms of spirituality (to link it with lines 90-95);
Response 5: We have linked lines 90-95 to the former introduction of spirituality, adding the meaningfulness of the GrAw-7 as a not religious questionnaire: Under the different definitions of spirituality, we have found the GrAw-7 (Gratitude/Awe ) which is to be an indicator of the experiential aspect of non-religious spirituality., and Tis thus it is applicable even for non-religious individuals. This aspect of spirituality is moderately to strongly related to various indicators of religious attitudes and practices (such as perception of the sacred) and intensity practices, but also to non-religious meditation practices and mindful awareness (Büssing, 2024; Büssing, Recchia, & Baumann, 2018; Büssing, Recchia, & Dienberg, 2018). This is an important aspect when investigating spiritual beliefs in European countries.
Comments 6: considering line 85, explain better the reasons for which South Tyrol is an Italian region with high vaccine hesitancy and it is important to evaluate the association spirituality and vaccine hesitancy in this geographical area;
Response 6: ).We have added the results from former studies: The effects of spirituality on vaccine hesitancy are investigated in the light of the cultural background of this region focusing even on interactions with conspiracy thinking and CAM (complementary and alternative medicine) use. Former investigations [Barbieri, 2024 und CAM studie einfügen], evidenced low vaccine uptake in South Tyrol, emphasizing lack of trust as one of the main reasons of vaccine hesitancy and evidencing the associations of vaccine hesitancy and the use of complementary and alternative medicine. Our interest was now to understand whether spirituality is linked to vaccine perceptions and may interact with these already known factors.
Comments 7: considering line 89, give the full form of GrAw-7;
Response 7: Done
Comments 8: in line 90 there is an unnecessary space "[...] non-religious spiritu-[...]";
considering lines 96-98, I cannot understand the reasons for which mental health was mentioned if it was wanted to evaluate spirituality and vaccine hesitancy. Please, explain better this choice and its implication in spirituality and vaccine hesitancy;
Response 8: We have added the text “Since we have found in [Barbieri, 2024] that vaccine hesitancy hase inclined in the years of the pandemic, were interested in understanding whether pandemic related vaccine perceptions and general vaccine perceptions were affected by spirituality and if yes, in the same way.”
Comments 9: reading line 117, there is only an inclusion criterion, but it seems to be unlikely. Please, define better inclusion and exclusion criteria for the participation of this study;
Response 9: We have tried to clarify : This study employed a cross-sectional probability-based mode survey. The Statistical Institute of the Autonomous Province of Bolzano-South Tyrol (ASTAT) recruited a random sample of fully aged citizens of South Tyrol, only excluding persons residing in nursing homes, utilizing a stratified sampling strategy by municipality, gender, and age group (18–34, 35–49, 50–64, 65+ years) with the program 'Surveyselect' in SAS v9.2 (SAS Institute Inc., Cary, North Carolina, United States).
Comments10: please, add the references of EU General Data Protection Regulation (line 118);
Response 10:
Comments 11: reading lines 129-134, I cannot understand when the questionnaire was administered (the period of data collection by the questionnaire) nor what the administration methods were (by mailing list? by telephone? by short message system?). Please, clarify this aspects;
Response 11: Added “Participants were invited via letter including the planned participation date, a link to the online questionnaire (with telephone support) covering demographic, clinical, and socio-behavioral aspects, and a personalized password for use as pseudo-anonymization code.” And the exact time period
Comment 12: being in South Tyrol, what was the language used for the questionnaire?
Response 12: added the text “The questionnaire was available in German and Italian language. The German and Italian versions, if not available from COSMO Italy and COSMO Germany surveys or as standardized questionnaire versions, were translated from ASTAT and reviewed for language equivalence by a research group at the Institute for General Practice and Public Health.”
Comment 13: reading lines 136-141, the point scales for the question group are confusing. I strongly suggest dividing the questions by point scales. For example, "Age, gender, educational level on a 4-point scale, [...]" is not clear: age can also have 4-point scale, but it must be defined better how the age classes are chosen, whereas gender can have more than 4-point scale. Whatever scale is chosen, it is important to provide evidence of why that modality was chosen rather than another, giving literature references when appropriate;
Response 13: We have changed the text: “Sociodemographic variables were used to predict vaccine agreement. Age in years, sex (male, female), educational level (middle school or less, vocational school, high school, university degree or more), Italian citizenship as a dichotomous variable, health profession as a dichotomous variable, chronic diseases as a dichotomous variable, and economic situation in the last three months (better, equal, worse, on a 3-point scale plus the option "do not know") were assessed.”
Comments 14: reading lines 139-141, explain better how the "Sociodemographic questions were adapted to the specific South Tyrolean contex";
Response 14: We have changed the text: “Sociodemographic, South Tyrol specific questions were added, including items for the municipality and for the mother tongue (German, Italian, Ladin, and others).”
Comment 15: reading lines 145-146, I cannot understand what number 2 and 3 of 4-point Likert scale corresponds to. The same is for the point Likert scales of the other questions. Please, clarify these aspects;
Response 15: All details added for the scale of lines 145-146. For the other scales, unfortunately, I cannot indicate the exact wording, since in the questionnaire we only indicated the first and the last one and the numbers between.
Comment 16: please, add the range of scores for each item (see lines 142-182);
Response 16: Done
Comment 17: reading line 148, there are unnecessary spaces in "[...] places / nature [...]";
Response 17: Done
Comment 18: as regards lines 179-180, questions about perceived necessity and harmfulness of vaccinations are lacking. Please, add the questions;
Response 18: Questions for perceived unnecessity regarding Covid-19 vaccination/mandatory childhood vaccination: Vaccination is not necessary, because…a. it is not effective; b. natural herd immunity and the immune system is quite enough; c. this disease does not/no longer exist; d. the whole thing is only a profit for the pharmaceutical industry. Questions for perceived harmfulness regarding Covid-19 vaccination/mandatory childhood vaccination: Vaccination is harmful, because… a. long-term risks are not known/risk are bigger than benefit; b. new vaccines pose additional risks in the RNA/not controlled enough; c. there are doctors who advise against it; d. compulsory corona vaccination with prioritization of certain groups will lead to major socio-political discussions/bad experiences
Comment 19: in line 195, Spearman's correlation coefficient is mentioned, but there is a reference about the interpretation. This reference Akoglu H. User's guide to correlation coefficients. Turk J Emerg Med. 2018 Aug 7;18(3):91-93. doi: 10.1016/j.tjem.2018.08.001. PMID: 30191186; PMCID: PMC6107969. can be helpful and added at the end of the sentence;
Response 19: done. Thank you!
Comment 20: please, use the acronym VIF after the full form of variance inflation factor (the acronym must be into brackets for the first time - line 205);
Response 20: Done
Comment 21: please, add the full form of the acronym DFBETA;
Response 21: Done
Comment 22: considering line 218, add the reference of the statistical software as it is reported in this website: https://www.ibm.com/support/pages/how-cite-ibm-spss-statistics-or-earlier-versions-spss;
Response 22: ok
Comment 23: after reading materials and methods section, I cannot understand how the variables were processed, what the independent variables are and the dependend variables. Please, clarify better;
Response 23: Thank you, you are right.Clarified this in the Statistics subsection: The sum scores of spirituality, perceived harmfulness of vaccination and perceived harmfulness of Covid-19 vaccination were regarded as outcomes/dependent variables. Further, changed the subsections of the methods into 2.2 demographics; 2.3: Spirituality; 3.4: Vaccinatin and vaccination perception; 3.5 Putative Predictors of Spirituality and Perceived harmfulness of vaccination (Independent Variables). Finally, added information about mediation analysis, as recommended later.
Comments 24: in line 224, there is a 22 into square brackets. Please, revise it;
Response 24: ok
Comments 25: reading lines 238-240, doses of COVID-19 vaccination are mentioned, but they are not included in the materials and methods section. The questions described in the materials and methods section has to be the same of those in the results section, otherwise the readers get confused. Please, revise this aspect paying more attention;
Response 25: ok, substituted doses with times, as mentioned in methos on lines 170-172
Comments 26: reading Supplementary Figure S1 (mentioned in line 245), I noted that there is not a relationship between variables, but a descriptive analysis (in detail, median, interquartile range, and outliers) depicted in the box plots. To have a relatioship between variables, a correlation test or regression should be performed, as it is reported in this article: Johannes Ledolter, Oliver W. Gramlich, Randy H. Kardon; Display of Data. Invest. Ophthalmol. Vis. Sci. 2020;61(6):25. https://doi.org/10.1167/iovs.61.6.25. For these reason, I strongly suggest revising the first part of the description of the Figure "Relationship [...] influenza vaccinations.". Moreover, this measure of central tendency is not mentioned in the statistical analysis section and in the last plot there is an incomplete word "Don't kn". Please, revise it;
Response 26: Changed the wording in the description and changed the graphic. Boxplots have been mentioned in the methods.
Comments 27: reading reading very carefully the results section (more than one), I noted that it is too hard to understand what results are significant for this reasearch. There are descriptions on how the variables were processed (which must be done in an appropriate section of materials and methods) and statistical analyses which are not reported in the relative section. To help in the revision of this section, I try to list how the results could be improved (I hope these comments are useful): - first of all, describe the participants' characteristics (total number of participant, age, gender, and items), paying attention with the acronym (for instance, standard deviation is SD); -
Response 27: The survey yielded a response rate of 36 percent, corresponding to 1,388 participants. Participants’ age had a mean+-standard deviation (SD) of 50.3+-17.48 and 51.0% of the participants were female. 18.1% had an educational status of middle school or less, 28.8% of vocational school, 31.4% of high school and 21.8% of university. 40.5% of the participants were urban residents and 90.8% had the Italian nationality. Regarding native language, 63.1% stated to speak German, 27.1% Italian, 3.7% Ladin and 6.1% other languages.
Comments 28: lines 227-230 should be inserted in other section which mentions the relationship between variables;
Response 28: Inserted into section 3.4
Comments 29: lines 231-232 should be put in the statistical analysis section;
Response 29: Done
Comments 30: lines 235-236 should be put in the appropriate section of materials and methods on sample methods;
Response 30: Done
Comments 31: lines 237-247 represent non-statistically significant results of Supplementary Figure S1, so they can be removed from the main manuscript and inserted in such Supplementary Material;
Response 31: Done
Comments32 : lines 253-255 should include statistically significant results for gender and nationality;
Response 32: Done
Comment 33: lines 259-261 can be removed because the results are not statistically results; -
Response 33: Done
Comments 34: line 262 mentions additional analysis, which are not reported in the statistical analysis section. Please, revise this aspect;
Response 34: Additional substituted by post hoc
Comments 35: lines 271-272 must report the statistically significant results (coefficents and other measures);
Response 35: Done
Comments 36: lines 273-276 should be removed because they do not report statistically significant results;
Response: 36: Done
Comments 37: - lines 277-285 should include only statistically significant results, removing those non-significant;
Response37: Done
Comments 38: - lines 286-296 report subgroups analysis considering economic status, which was described in the statistical analysis section. Please, before reporting the results, describe the analysis in the statistical analysis section. Moreover, report only the statistically significant results. Do the same for the lines 297-306;
Response 38: done
Comments 39: lines 324-325 should be removed;
Response 39: Done
Comments 40: lines 327-344 should include only the statistically significant results;
Response 40: Done
Comments 41: lines 349-353 should be included in the statistical analysis section;
Response 41: Done
Comments 42: lines 353-358 should be included in the appropriate subsection of the materials and methods;
Response 42: Done
Comments 43: lines 359-360 should be included in the statistical analysis section;
Response 43: Done
Comments 44: lines 374-376 and 393-396 should be included in the statistical analysis section; the subsection
Response 44: I’m sorry, but this comment is not clear for me. Why should I put the model diagnostics results in the statistical analysis part?
Comments 45: "3.5.1. Mediation Model Overview" should be included in the statistical analysis section;
Response 45: Done
Comments 46: line 514 reports an unnecessary bracket;
Response 46: I’m sorry, I did not find the bracket.
Comments 47: lines 549-551 seem to be without references. Please, revise them;
Response 47: Following text and references were added: “Research indicates a modest correlation between spirituality and conspiracy thinking, both of which can exacerbate vaccine hesitancy. For instance, a study found that individuals with holistic spiritual beliefs are more susceptible to conspiracy theories, which in turn can lead to skepticism toward vaccinations (Jedinger & Siegers, 2024). Similarly, research has shown that lower health literacy and higher religiosity are associated with increased belief in vaccination-related conspiracies (Pavić et al., 2023).”
Comments 48: considering the paragraph "4.4. Strengths and Limitations", common biases for the cross-sectional study must be reported;
Response 48: changed into: “. First, its cross-sectional design precludes longitudinal interpretations, while associations between spirituality and vaccine perceptions were identified but cannot be interpreted in the sense of causal inference; it is not feasible to infer how they change and affect each other over time.”
Comments 49: lines 669-672 do not report an approval by a local ethic committee in accordance with the Legislative Decree no. 121 of May 5, 2001. Searching it in the Internet, this source is not available and I cannot check the corrispondence between the Legislative Decree and the Authors' declaration. Although I trust them, I would like to check it. For this reason, I would like to ask the link of the source.
Response 49:
The survey is included in the state statistics programme 2024- 2026 (BSL 040), which was approved by resolution of the state government (No. 43/2024). The information collected as part of this study is protected by the Italian Statistics Act (Art. 9, Legislative Decree no. 322/1989) and is subject to the provisions on the protection of personal data (EU Regulation 679/2016 and Legislative Decree no. 196/2003 as last amended by Legislative Decree no. 101 of 10 August
2018).
We attach the informed consent for the participants (Consenso informato_Health Literacy_en-it en.pdf). I hope this can help

Round 2
Reviewer 6 Report
Comments and Suggestions for Authors
I would like to thank the Authors for addressing my comments and I congratulate them for their efforts in improving the manuscript quality. I would have some comments:
- line 16: there is a space between the word and the bracket "[...] medicine ) use on [...]";
- line 26: there is a suspicious verb at beginning of the sentence "[...] was All three variables [...];
- line 86: check the space "[...] hesitancy. South Tyrol, [...]";
- line 109: there are double t at the beginning of the sentence "Tthis study [...]";
- the text into lines 119-122 is very clear, but it should be considered removing the text 136-140 (less clear than the before). Please, check it;
- line 153: put one space between a word and the bracket "[...] educational level(middle school [...]";
- line 157: there is an unnecessary space "[...] questions were added [...]";
- line 162: put a comma and then the number 4 before "regularly/very often";
- line 235: there is an unnecessary space "[...] predictors, were incorporated [...]";
- reading better Table 1, I suggest that it can be laid out horizontally, so there is more space to insert the effect size and its p-value next to the Spearman coefficient;
- considering lines 288-294, Cronbach’s alpha has to be mentioned as a statistical analysis conducted in this study in the statistical analysis section. To have a range of reliability, see this article: Mohd Arof, Khairul & Ismail, Syuhaida & Saleh, Abd Latif. (2018). Contractor’s Performance Appraisal System in the Malaysian Construction Industry: Current Practice, Perception and Understanding. International Journal of Engineering & Technology. 7. 46. 10.14419/ijet.v7i3.9.15272.;
- line 348: I cannot understand the meaning of VB2. Please, clarify;
- line 378: a space is needed "[...] COVID-19partici [...]";
- reading Table 2, check unnecessary and lacking spaces between numbers and brackets;
- line 473: put a bracket after Table 4;
- line 523, 553, and 619: there are suspicious words "[...] VB3][CW4 [...]", "[...] [VB5][CW6] [...]", "[...] VB10". Please, clarify;
- 4.4. Strengths and Limitations: how were biases (for example, social desirability bias or missing bias) dealt with?
- Supplementary material S2: Mann-Whitney U and Kruskal-Wallis tests have to be mentioned and for variables they were used in the statistical analysis section.
Regarding the comment 44, I am sorry for my misunderstanding.
After addressing these comments, the manuscript can be published. I would like to send my best wishes to the Authors for their future research in this field.
Author Response
@Reviewer 6: Thank you for your comments. I was impressed on how much time and interest you spent to our paper. I've really learned a lot.
Comments 1: line 16: there is a space between the word and the bracket "[...] medicine ) use on [...]";
Response 1: ok
Comments 2: line 26: there is a suspicious verb at beginning of the sentence "[...] was All three variables [...];
Response 2: ok
Comments 3: line 86: check the space "[...] hesitancy. South Tyrol, [...]";
Response 3: ok
Comments 4: line 109: there are double t at the beginning of the sentence "Tthis study [...]";
Response 4: ok
Comments 5: the text into lines 119-122 is very clear, but it should be considered removing the text 136-140 (less clear than the before). Please, check it;
Response 5: ok
Comments 6: line 153: put one space between a word and the bracket "[...] educational level(middle school [...]";
Response 6: ok
Comments 7: line 157: there is an unnecessary space "[...] questions were added [...]";
Response 7: ok
Comments 8: line 162: put a comma and then the number 4 before "regularly/very often";
Response 8: ok
Comments 9: line 235: there is an unnecessary space "[...] predictors, were incorporated [...]";
Response 9: ok
Comments 10: reading better Table 1, I suggest that it can be laid out horizontally, so there is more space to insert the effect size and its p-value next to the Spearman coefficient;
Response 10: done
Comments 11: considering lines 288-294, Cronbach’s alpha has to be mentioned as a statistical analysis conducted in this study in the statistical analysis section. To have a range of reliability, see this article: Mohd Arof, Khairul & Ismail, Syuhaida & Saleh, Abd Latif. (2018). Contractor’s Performance Appraisal System in the Malaysian Construction Industry: Current Practice, Perception and Understanding. International Journal of Engineering & Technology. 7. 46. 10.14419/ijet.v7i3.9.15272.;
Response 11: In Methods added the text “According to Arof et al. [34][], Cronbach’s alpha of more than 0.9 was regarded as excellent, of 0.8-0.89 as good and of 0.7-0.79 as acceptable.”
Comment 12: line 348: I cannot understand the meaning of VB2. Please, clarify;
Response 12: Removed Comments
Comments 13: line 378: a space is needed "[...] COVID-19partici [...]";
Response 13: ok
Comment 14: reading Table 2, check unnecessary and lacking spaces between numbers and brackets;
Response 14: Removed
Comments 15: line 473: put a bracket after Table 4;
Response 15: ok
Comments 16: line 523, 553, and 619: there are suspicious words "[...] VB3][CW4 [...]", "[...] [VB5][CW6] [...]", "[...] VB10". Please, clarify;
Response 16: Removed Comments
Comments: 17: 4.4. Strengths and Limitations: how were biases (for example, social desirability bias or missing bias) dealt with?
Response 17: Added text “Finally, the sampling method did not allow participants to partially answer the questions. Thus, we were able to evaluate only complete questionnaires. Post-stratification weights adjusted for non-response bias. Social desirability bias was accounted for by avoiding sentences having emotionally positive or negative connotations as well as questions that are similar to known slogans or have a desirable or undesirable response. Further, socially accepted, or unaccepted wording was avoided.”
Comments 18: Supplementary material S2: Mann-Whitney U and Kruskal-Wallis tests have to be mentioned and for variables they were used in the statistical analysis section.
Response 18: Added variables in the statistical analysis section: For more than two groups (educational level, mother tongue, economic status, opinion change regarding vaccination, agreement with flu vaccination, agreement with childhood vaccination, times of Covid-19 vaccination) Kruskal-Wallis tests were employed